# WetCH4: A Machine Learning-based Upscaling of Methane Fluxes of Northern Wetlands during 2016-2022

Qing Ying[1,2*], Benjamin Poulter[2], Jennifer D. Watts[3], Kyle A. Arndt[3], Anna-Maria Virkkala[3], Lori Bruhwiler[4], Youmi Oh[4,5], Brendan M. Rogers[3], Susan M. Natali[3], Hilary Sullivan[3], Amanda Armstrong[2], Eric J. Ward[2], Luke D. Schiferl[6], Clayton D. Elder[7,8], Olli Peltola[9], Annett Bartsch[10], Ankur R. Desai[11], Eugénie Euskirchen[12], Mathias Göckede[13], Bernhard Lehner[14], Mats B. Nilsson[15], Matthias Peichl[15], Oliver Sonnentag[16], Eeva-Stiina Tuittila[17], Torsten Sachs[18,19], Aram Kalhori[18], Masahito Ueyama[20], Zhen Zhang[1,21*]

[1]Earth System Science Interdisciplinary Center, University of Maryland, College Park, MD 20740, USA
[2]Biospheric Sciences Laboratory, NASA Goddard Space Flight Center, Greenbelt, MD 20771, USA
[3]Woodwell Climate Research Center, Falmouth, MA 02540, USA
[4]NOAA ESRL, 325 Broadway, Boulder, Colorado, USA
[5]Cooperative Institute for Research in Environmental Sciences (CIRES), University of Colorado, Boulder, CO 80305, USA
[6]Lamont-Doherty Earth Observatory, Columbia University, Palisades, NY, USA
[7]Jet Propulsion Laboratory, California Institute of Technology, Pasadena, CA, USA
[8]Ames Research Center, Earth Science Division, Moffett Field, CA, USA
[9]Natural Resources Institute Finland (Luke), Latokartanonkaari 9, Helsinki, 00790, Finland
[10]b.geos, Korneuburg, Austria
[11]Department of Atmospheric and Oceanic Sciences, University of Wisconsin, Madison, WI 53706, USA
[12]Institute of Arctic Biology, University of Alaska, Fairbanks, AK 99775, USA
[13]Max Planck Institute for Biogeochemistry, Jena, Germany
[14]Department of Geography, McGill University, Montreal, QC H3A 0B9, Canada
[15]Department of Forest Ecology and Management, Swedish University of Agricultural Sciences, Umeå, Sweden
[16]Département de géographie, Université de Montréal, Montréal, QC, Canada
[17]University of Eastern Finland, School of Forest Sciences, Joensuu, Finland
[18]GFZ Helmholtz Centre for Geosciences, Potsdam, Germany
[19]Institute of Geoecology, Technische Universität Braunschweig, Braunschweig, Germany
[20]Graduate School of Agriculture, Osaka Metropolitan University, Japan
[21]National Tibetan Plateau Data Center (TPDC), State Key Laboratory of Tibetan Plateau Earth System, Environment and Resource (TPESER), Institute of Tibetan Plateau Research, Chinese Academy of Sciences, Beijing, 100101, China

*Correspondence to*: Qing Ying ([qying@umd.edu](mailto:qying@umd.edu)) and Zhen Zhang ([yuisheng@gmail.com](mailto:yuisheng@gmail.com))

# 42 Abstract

Wetlands are the largest natural source of methane ($CH_4$) emissions globally. Northern wetlands
(>45° N), accounting for 42% of global wetland area, are increasingly vulnerable to carbon loss,
especially as $CH_4$ emissions may accelerate under intensified high-latitude warming. However,
the magnitude and spatial patterns of high-latitude $CH_4$ emissions remain relatively uncertain.
Here we present estimates of daily $CH_4$ fluxes obtained using a new machine learning-based
wetland $CH_4$ upscaling framework (WetCH$_4$) that combines the most complete database of eddy
covariance (EC) observations available to date with satellite remote sensing informed
observations of environmental conditions at 10-km resolution. The most important predictor
variables included near-surface soil temperatures (top 40 cm), vegetation spectral reflectance,
and soil moisture. Our results, modeled from 138 site-years across 26 sites, had relatively
strong predictive skill with a mean $R^2$ of 0.51 and 0.70 and a mean absolute error (MAE) of 30
nmol m$^{-2}$ s$^{-1}$ and 27 nmol m$^{-2}$ s$^{-1}$ for daily and monthly fluxes, respectively. Based on the model
results, we estimated an annual average of 22.8 ±2.4 Tg $CH_4$ yr$^{-1}$ for the northern wetland
region (2016-2022) and total budgets ranged from 15.7 - 51.6 Tg $CH_4$ yr$^{-1}$, depending on
wetland map extents. Although 88% of the estimated $CH_4$ budget occurred during the May-
October period, a considerable amount (2.6 ±0.3 Tg $CH_4$) occurred during winter. Regionally,
the West Siberian wetlands accounted for a majority (51%) of the interannual variation in
domain $CH_4$ emissions. Overall, our results provide valuable new high spatiotemporal
information on the wetland emissions in the high-latitude carbon cycle. However, many key
uncertainties remain, including those driven by wetland extent maps and soil moisture products,
incomplete spatial and temporal representativeness in the existing $CH_4$ flux database – e.g.,
only 23% of the sites operate outside of summer months and flux towers do not exist or are
greatly limited in many wetland regions. These uncertainties will need to be addressed by the
science community to remove bottlenecks currently limiting progress in $CH_4$ detection and
monitoring. The dataset can be found at [https://doi.org/10.5281/zenodo.10802153](https://doi.org/10.5281/zenodo.10802153) (Ying et al.,
68  2024).


Keywords
Northern high latitudes; wetland; methane ($CH_4$) flux; eddy covariance; remote sensing;
machine learning; data-driven upscaling

# 73 1. Introduction

Methane ($CH_4$) is the second most important greenhouse gas after carbon dioxide ($CO_2$) and
has contributed to around 1/3 of anthropogenic warming (IPCC AR6, 2023). Wetlands are the
largest natural source of $CH_4$ emissions. Northern freshwater wetlands (>45° N) account for
roughly 40% of global wetland area (ranging 1.3 - 8.7 million km$^2$; (Zhang et al., 2021)), yet the
amount of $CH_4$ emissions from this region is highly uncertain – currently estimated to be 22 –
49.5 Tg $CH_4$ yr$^{-1}$ (Aydin et al., 2011; Baray et al., 2021; Heimann, 2011; Kirschke et al., 2013;
Peltola et al., 2019; Saunois et al., 2020; Treat et al., 2018; Watts et al., 2023). The
uncertainties in the estimates of wetland $CH_4$ emissions are primarily attributed to challenges in
mapping vegetated wetlands versus open water leading to double counting (Thornton et al.,
2016), seasonal wetland dynamics and uncertainties in estimates on flux rates.
Characterized by nutrient, moisture and hydrodynamic conditions, northern freshwater wetlands
are classified as wet tundra in treeless permafrost areas, peat-forming bogs and fens in boreal
and temperate biomes, and permafrost bogs (Kuhn et al., 2021; Olefeldt et al., 2021). Bogs
were estimated to cover the largest area (1.38-2.41 million $km^2$) in the northern high latitudes,
followed by fens (0.76-1.14 million $km^2$) and wet tundra (0.31-0.53 million $km^2$) (Olefeldt et al.,
2021). Climate change poses significant threats to these wetlands, affecting their extent and the
duration of conditions suitable for wetland formation in permafrost zones (Avis et al., 2011). The
rates of $CH_4$ emissions may increase quickly because of intensified warming at the northern
high latitudes (Masson-Delmotte et al., 2021; Rawlins et al., 2010; Rößger et al., 2022; Walsh,
2014; Zhang et al., 2023).
Reflecting CH4 response to warming, northern wetlands may account for a high portion
(~78.5%) of the global surface emissions anomaly of $CH_4$ in 2020 relative to 2019 (6.0 ± 2.3 Tg
$CH_4$ $yr^{-1}$) (Peng et al., 2022; Zhang et al., 2023). This is concerning as the responses of high
latitude $CH_4$ emissions to a warming and possibly wetting climate could produce a positive
carbon-climate feedback (McGuire et al., 2009; Natali et al., 2019). However, the ability of
models to account for and predict the spatio-temporal variability of high latitude wetland $CH_4$
emission rates remain very limited (Treat et al., 2024).
Field observations of gas fluxes typically measure $CH_4$ exchange between the land and
atmosphere at sub-meter to ecosystem (100s of m to km) scales (Bansal et al., 2023; Chu et al.,
2021). Tower eddy covariance (EC) methods provide near-continuous measurements over
ecosystem-scale footprints (5 – 100 x $10^3$ $m^2$), the size of which matches fine to medium
resolution satellite remote sensing. Typical EC measurement system records include carbon,
water and energy fluxes along with environmental conditions at half-hourly intervals. Long-term
EC datasets can support the analysis of daily, monthly, seasonal, or interannual patterns and
drivers of carbon fluxes (Baldocchi, 2003). Chambers can also measure $CH_4$ fluxes, though at
sub-meter resolution and small spatial coverage area (Bansal et al., 2023; Kuhn et al., 2021).
Most chamber studies have a limited temporal sampling period. To avoid footprint disagreement
between EC and chamber measurement techniques, we focused on EC-based $CH_4$ upscaling in
this study.
Data-driven upscaling uses empirical models (Bodesheim et al., 2018; Jung et al., 2011),
including machine learning (ML) approaches, to compute $CH_4$ fluxes. It provides independent
estimates to those from process-based models and atmospheric inversions (Bergamaschi et al.,
2013; Spahni et al., 2011). These approaches have been used to estimate $CH_4$ fluxes from
various ecosystems such as northern wetlands (Peltola et al., 2019; Virkkala et al., 2024; Yuan
et al., 2024), global reservoirs (Johnson et al., 2021), and global aquatic ecosystems
(Rosentreter et al., 2021).

Two types of methods are generally used for data-driven upscaling. The first uses a look-up
table approach and applies emission rates or emission factors via data synthesis to the
corresponding land surface areas, or activity data, over the study region. Emission rates from
field observations are associated with environmental drivers that have been spatially
characterized and are then applied to the land covers with the same environmental drivers. For
example, Rosentreter et al. (2021) collected 2,601 $CH_4$ flux records measured using various
methods through a literature review and characterized emission rates over 15 aquatic
ecosystem types to upscale global aquatic $CH_4$ emissions. The study provided estimates of total
and per ecosystem emissions but did not produce spatial distributions and was unable to
estimate temporal changes. A similar method was applied for the northern permafrost region,
where statistical $CH_4$ flux rates from the Boreal-Arctic Wetland and Lake $CH_4$ Dataset (BAWLD-
$CH_4$) were analyzed for emission estimation by wetland type (Kuhn et al., 2021; Ramage et al.,
2024). This method favors homogeneous ecosystems and static environments, and the results
may be biased for large-scale studies where spatial heterogeneity is prevalent.

Another approach uses ML methods to upscale fluxes (Bodesheim et al., 2018; Tramontana et
al., 2016; Yuan et al., 2024). ML models are developed with large training datasets. Generally,
ML models can learn from high-dimensional data by optimizing many statistical parameters and
identifying variables that are closely associated with spatio-temporally varied $CH_4$ emissions.
The efficient computation cost makes it easier to apply the models over large regions at higher
spatial resolutions. Among ML methods, decision-tree-based algorithms have been widely used
in upscaling for computation efficiency and prediction performance (Beaulieu et al., 2020; Jung
et al., 2020; Virkkala et al., 2021; Zhang et al., 2020). Specifically, Random Forests (RF) were
utilized in regional to global wetland $CH_4$ upscaling (Davidson et al., 2017; Feron et al., 2024;
McNicol et al., 2023; Peltola et al., 2019) for the robustness and prevention of overfitting to
noise in the input data. For example, Peltola et al. (2019) used RF and EC measurements to
upscale monthly $CH_4$ fluxes from the northern wetlands at 0.25°- 0.5° spatial resolution over the
2013-2014 period.

ML-based upscaling studies usually incorporate information from remote sensing to inform
wetland extent, changes in vegetation and other surface biophysical properties (Davidson et al.,
2017; Virkkala et al., 2024; Watts et al., 2014, 2023). For example, recent ML-based large-scale
upscaling approaches used MODIS land surface temperature at night (LST), enhanced
vegetation index (EVI), vegetation canopy height, and ancillary environmental variables from
remote sensing products (McNicol et al., 2023; Ouyang et al., 2023; Peltola et al., 2019) (See
Supporting Materials S1 and Table S1 for detailed predicting variables used in existing ML-
based wetland $CH_4$ upscaling products). However, soil moisture and soil temperature, two
controlling factors of wetland $CH_4$ fluxes (Knox et al., 2021; Yuan et al., 2022), were missing in
previous ML-based regional to global upscaling studies. Soil moisture has been identified as
one of the important controlling factors for freshwater wetland $CH_4$ fluxes (Euskirchen et al.,
2024; Voigt et al., 2023). This is the first ML-based study that incorporates remote sensing
constraints from Soil Moisture Active Passive (SMAP) microwave-sensed soil moisture and
Moderate Resolution Imaging Spectroradiometer Nadir Bidirectional Reflectance Distribution
Function (BRDF) – Adjusted Reflectance (MODIS NBAR) data. Surface reflectance provides
information of vegetation properties that affect the production and transport of $CH_4$ to the
atmosphere, and ecosystem wetness (Alonso et al., 2020; Chen et al., 2013; Entekhabi et al.,
2010; Houborg et al., 2007; Murray-Hudson et al., 2015; Wang et al., 2018).
The goal of this study is to develop a scalable framework to upscale daily $CH_4$ fluxes from EC
observations to northern latitude wetlands (>45° N) using the ensemble RF ML approach with a
suite of reanalysis and remote sensing products representing spatiotemporal environmental
conditions. Our specific objectives are to:
1. compile an updated EC-based $CH_4$ flux dataset that extends the temporal and spatial
coverage of the Fluxnet-$CH_4$ database (Delwiche et al., 2021) for the northern high
latitudes;
2. build RF models of $CH_4$ fluxes at site-level based on *in-situ* measured physical variables
which allow us to prioritize the selection of gridded variables for upscaling, and then
build ensemble RF models at grid-level using gridded reanalysis inputs and constraints
from satellite remote sensing; and
3. apply grid-level models to produce a 10-km gridded daily distribution of $CH_4$ flux product
for the northern high latitude wetlands using bootstrapped models and their derived
uncertainties (Table S1).

# 187 2. Materials and methods

## 188 2.1 Overview

The scalable framework of upscaling $CH_4$ fluxes from EC observations for wetlands (referred to
as WetCH4 hereafter), which selects physical predictors at the site level and constructs
upscaling models at a grid level, is illustrated in Fig. 1. *In situ*, reanalysis, and remote-sensing
products were compiled as candidate predictors for modeling (Fig. 1, purple boxes; see section
2.2 for details). We first ran a feature selection, which uses ensemble RF models to choose
important predictors from an extensive list of *in situ* variables available from the flux tower sites.
Gridded versions of selected site variables were taken from Modern-Era Retrospective analysis
for Research and Applications (MERRA2) reanalysis (Gelaro et al., 2017) to model with RF at
grid level. We then added remote sensing-based products from MODIS NBAR and SMAP soil
wetness, as well as topographic data, to strengthen the model and provide finer delineation of
environment gradients based on literature and expert knowledge (Poulter et al., 2023;
Sturtevant et al., 2012). The predictive performance of grid-level models with input variables at
their native spatial resolution (except for MERRA2 that were interpolated to 10-km resolution)
was then evaluated. We also compared model performance with those from two additional ML
algorithms: support vector machines (SVM) and artificial neural network (ANN) (Fig. 1 pink
boxes). The RF algorithm modeled on all gridded input variables gained the highest mean $R^2$
and lowest daily median errors in model predictive performance and was selected for bootstrap
modeling and upscaling the 0.098° (~10km along latitudinal length) gridded time series of daily
$CH_4$ fluxes and ensemble uncertainty estimation (Fig. 1 grey boxes).

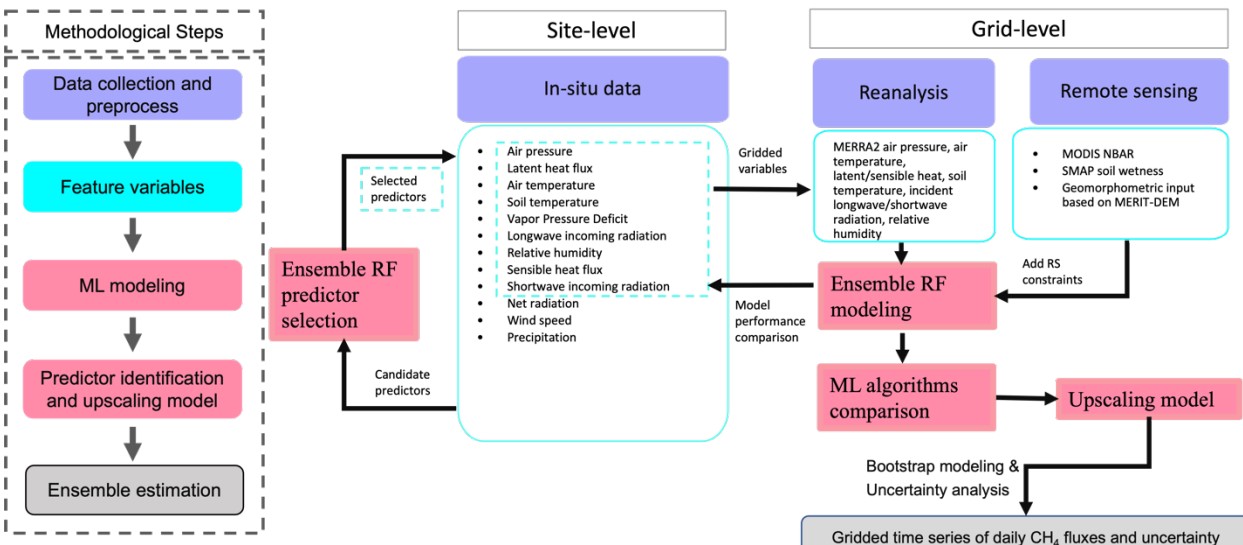

Figure 1 Workflow and experimental design: abstract methodological steps are integrated in the
dashed box on the left, while a detailed experimental design is described on the right. Colors on
the right match the associated step on the left.

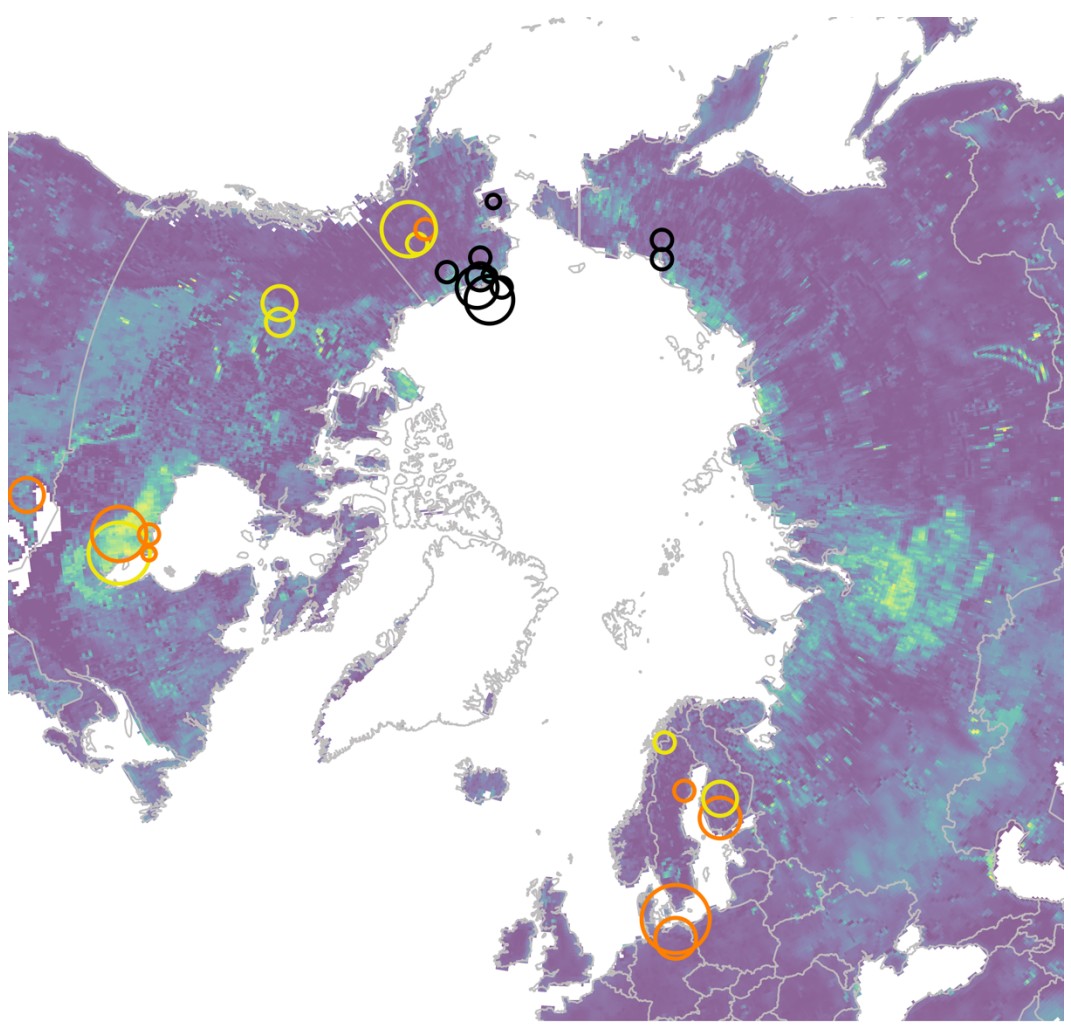

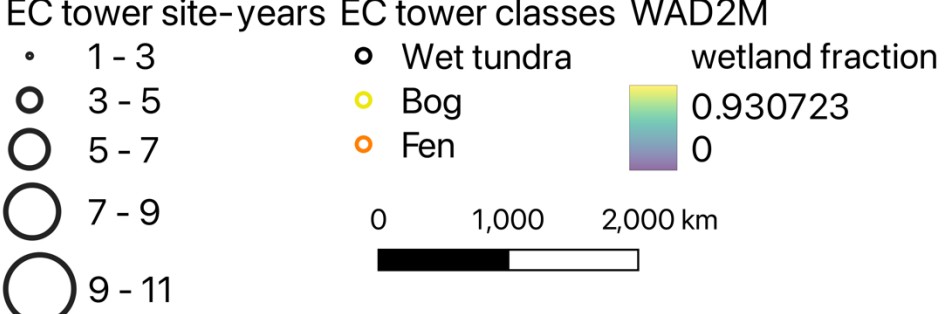

Figure 2 Eddy covariance tower sites: distribution (>45° N), class, and data size (site-years)
used in WetCH$_4$. Colored circles represent EC tower locations and land cover classes, with
wetland sites in cyan (wet tundra), yellow (bog) and orange (fen). The circle sizes represent
observation years(n) of available CH$_4$ fluxes at the site (e.g. 1-3 stands for 1<=n<3). The
background image shows the estimated maximum annual fractions of wetland cover in 2011-
2020 from WAD2M (Zhang et al., 2021).

## 2.2 Data

### 2.2.1 Eddy covariance measurements

The base of our EC data collection stems from a publicly available global synthesis coordination of FLUXNET-CH$_4$ (Delwiche et al., 2021; Knox et al., 2019), which includes 79 EC tower sites (42 are freshwater wetland sites) and 293 site-years of data. Fluxnet-CH$_4$ represents a first compilation of global CH$_4$ fluxes measured by EC towers (Delwiche et al., 2021; Knox et al., 2019), yet more EC data exists outside of the network. We collected both daily and half-hourly data from 44 sites in the northern high latitudes (>45° N), accounting for 167 site years as our base dataset, to which we added data from 6 new sites (31 site-years) and added additional data to 9 existing sites (21 site-years) contributed by principal investigators (Table S2). In total, we assembled data from 50 EC tower sites in northern latitudes (219 site-years), of which 33 are from wetlands (155 site-years), with 13 wet tundra sites, 11 fens, and 9 bogs. Data entries with missing data in gridded predictors were excluded, including 5 wetland sites (FI-LOM, DE-SFN, RU-SAM, RU-VRK, SE-ST1) where data was collected before SMAP data was available. Another 2 sites (CA-BOU, RU-COK) were excluded after quality control revealed an instrument anomaly that affected the measurements. As a result, daily and half-hourly EC data from the 26 wetland sites were compiled for analysis from 22 sites in FLUXNET-CH$_4$ (among which 8 sites with updated data to recent years including US-ATQ, US-BEO, US-BES, US-BRW, US-IVO, US-NGB, US-NGC, US-UAF) and 4 additional sites using information provided directly by principal investigators (including CA-ARB, CA-ARF, CA-PB1, CA-PB2), consisting of 138 site-years data in total and representing the largest high latitude EC-data compilation for CH$_4$ to date (Table S2, see Supporting Materials S2). The sites were distributed among wetland types, including 9 fens, 7 bogs, and 10 wet tundra sites (Fig. 2). RU-CHE and RU-CH2 were two Chersky sites in East Siberian Russia about 600m apart from each other to form a paired disturbance experiment. RU-CH2 was a control tower over an undisturbed wetland, whereas RU-CHE was a tower affected by artificial drainage. The above-ground conditions of the two sites were virtually identical, but soil temperature and moisture were different. Drainage caused lower CH$_4$ fluxes at RU-CHE compared to those at RU-CH2. However, the SMAP data could not discern the drainage impact on soil moisture at the RU-CHE site due to a coarser spatial resolution, thus it was excluded from grid-level modeling.

Half-hourly fluxes acquired from FLUXNET-CH$_4$ were already gap-filled (see Supporting Materials S2; (Irvin et al., 2021)). Additional half-hourly fluxes acquired from site PIs were not gap-filled, and as such we performed per site gap filling following the FLUXNET-CH$_4$ approach (Irvin et al., 2021; Knox et al., 2019). Gap-filled fluxes were temporally consistent and agreed with validation data (mean $R^2$ = 0.68 and mean RMSE = 6 nmol m$^{-2}$ s$^{-1}$, *see* Supporting Materials S2).

The mean difference in daily mean fluxes between the gap-filled data and the original data converged to -0.2 nmol m$^{-2}$ s$^{-1}$ when there were more than 11 half-hourly EC tower observations in a day but showed substantial bias and larger differences when including days with less than 11 half-hourly observations (Fig. S1). Therefore, daily data entries were only kept when the

number of half-hourly EC tower observations per day was greater than 11. All data were
retained on four sites where only daily, quality-filtered, data were provided by site PIs (Table
S2). As a result, we identified 12,784 daily data entries for upscaling models (Table
S2), spanning 2015-2021 with seasonal observation distributions of 44.0% in June-July-August
(JJA), 29.0% in March-April-May (MAM), 24.5% in September-October-November (SON), and
2.5% in December-January-February (DJF) (Fig. S2).
Site-level candidate predictors were identified by their known influences on $CH_4$ fluxes at multi-
day to seasonal scales from field control experiments, *in situ* flux synthesis, and process-based
modeling (Bloom et al., 2010, 2017; Knox et al., 2021; Olefeldt et al., 2013, 2017). Only in situ
measured physical variables were considered candidate predictors at site-level modeling. *In situ*
candidate predictors that were gap-filled and available in FLUXNET-$CH_4$ included daily
averages of air temperature, soil temperature, air pressure, vapor pressure deficit, relative
humidity, latent heat flux, sensible heat flux, longwave incoming radiation, shortwave incoming
radiation, net radiation, wind speed, and daily total precipitation (Fig. 1 site-level model solid
blue box). We were unable to include water-table depth (WTD) or soil water content (SWC) in
our site-level model as they were not available at many sites. However, we explored ML results
that included WTD or SWC for a subset of individual sites (36% of total) where these variables
were available (see Supporting Materials S2 for more details).

## 2.2.2 Reanalysis data and satellite data products
Reanalysis data were used as the gridded input to replace selected predictors at the site level
for training the grid-level models and upscaling. These data provided long-term continuous
estimates of nearly all the candidate predictors of the *in situ* measured variables (Fig. 1).
MERRA2 is an atmospheric reanalysis of the modern satellite era produced by NASA's Global
Modeling and Assimilation Office (Gelaro et al., 2017). We calculated daily means for air
pressure, surface air temperature, latent heat flux, sensible heat flux, downward-incoming
shortwave radiation, downward-incoming longwave radiation, and soil temperature at three
depths (9.88 cm, 19.52 cm, 38.59 cm) (Jiao et al., 2023), and relative humidity using the hourly
average of surface flux diagnostics, land surface diagnostics, and land surface forcings. The
original 0.5° x 0.625° resolution data were resampled to 0.5° grids using a bilinear interpolation
method in the NASA MERRA2 web service tool available on GES DISC. The MERRA2 data
were further bilinearly interpolated from 0.5° to 0.098° grids weighted by the multiple-error-
removed improved-terrain digital elevation model (MERIT-DEM) at 90-m resolution that
significantly improves elevation estimates in flat terrain over previous DEM products (Yamazaki
et al., 2017). Daily time series of the nearest 0.098° grid to each EC location were extracted for
grid-level modeling, whereas daily grids were input for the 10-km upscaling products.
To improve the predictive performance of grid-level models, we added remotely sensed
biophysical variables, including SMAP soil wetness, MODIS NBAR bands, and topographic data
(Fig. 1, Table 1). All remote-sensing products were extracted in daily time steps and their native
spatial resolutions at EC tower sites for modeling and aggregated to 0.098° grids over the study
domain for upscaling using Google Earth Engine. We filtered out data gaps in SMAP and
MODIS NBAR time series extracted at the native spatial resolution during model training and
validation. Gaps in MODIS NBAR were negligible when aggregated from 500-m to 0.098° grids.
Gaps in winter SMAP data were filled with zero values to represent frozen soils for upscaling.
The SMAP soil moisture product is generated using passive microwave radiometer-measured
brightness temperature merged with estimates from the GEOS Catchment Land Surface and
Microwave Radiative Transfer Model in a soil moisture data assimilation system, providing
global products of surface and rootzone soil moisture (Reichle et al., 2017). For soil moisture,
we employed Level-4 daily soil wetness products (SPL4SMGP.007) from the SMAP mission as
proxies for water-table depth in the grid-level model (Reichle et al., 2017). Surface, rootzone,
and soil profile wetness are dimensionless variables that indicate relative saturation for top layer
soil (0-5 cm), root zone soil (0-100 cm), and total profile soil (0 cm to model bedrock depth),
respectively. These three variables are originally 3-hourly data at 9-km resolution and were
converted to daily means.
Static topographic variables were added as additional attributes in the grid-level model. We
used topographical slope and indices that represent the water flow from MERIT-DEM based on
Geomorpho90m (Amatulli et al., 2020). Two topographic indices were applied: the compound
topographic index (cti) is considered a proxy of the long-term soil moisture availability, and the
stream power index (spi, https://gee-community-catalog.org/projects/geomorpho90/) reflects the
erosive power of the flow and the tendency of gravitational forces to move water downstream.
We tested the impact of elevation on model performance in explaining inter-site variability of
$CH_4$ upon the current locations of wetland EC sites (see Supporting Materials S6).
Nevertheless, elevation was not considered an ecologically controlling factor for wetland $CH_4$
fluxes, and hence was excluded from the input variable importance analysis that ranked the
importance of predictors to the prediction accuracy in RF models.
We included MODIS NBAR (MCD43A4v061) products as predictor variables to represent the
vegetation productivity in the grid-level model in order to enhance our model predictive
performance in vegetated wetlands. The 7 NBAR bands (including red/green/blue, 2 near
infrared, and 2 shortwave infrared) are developed daily at 500-m spatial resolution, using 16
days of Terra and Aqua data to remove view angle effects, and it is temporally weighted to the
ninth day as the best local solar noon reflectance (Schaaf et al., 2002; Wang et al., 2018). We
did not explicitly include a vegetation productivity variable, because such information is retained
in MODIS NBAR that is used to produce vegetation indices (e.g. EVI) and gross primary
production (GPP). Emergent aerenchymous vegetation is another important component in the
plant-mediated pathway of $CH_4$ transport yet was less represented in existing upscaling models
(Table S1).
Table 1. Description of input variables for grid-level upscaling model

| Variable type | Name | Description | Unit | Data source | Native/Model Spatial resolution | Native Temporal |
|---|---|---|---|---|---|---|

|  |  |  |  |  |  | resolution |
| --- | --- | --- | --- | --- | --- | --- |
| Reanalysis | tas | surface air temperature | °C | MERRA2 | 0.625°×0.5°/10km | 1 hourly |
| Reanalysis | pa | surface air pressure | Kpa | MERRA2 | 0.625°×0.5°/10km | 1 hourly |
| Reanalysis | le | latent heat | W m$^{-2}$ | MERRA2 | 0.625°×0.5°/10km | 1 hourly |
| Reanalysis | h | sensible heat | W m$^{-2}$ | MERRA2 | 0.625°×0.5°/10km | 1 hourly |
| Reanalysis | rsdl | downward-incoming longwave radiation | W m$^{-2}$ | MERRA2 | 0.625°×0.5°/10km | 1 hourly |
| Reanalysis | rsds | downward-incoming shortwave radiation | W m$^{-2}$ | MERRA2 | 0.625°×0.5°/10km | 1 hourly |
| Reanalysis | spfh | surface specific humidity | unitless | MERRA2 | 0.625°×0.5°/10km | 1 hourly |
| Reanalysis | ts1 | soil temperature | ° C | MERRA2 | 0.625°×0.5°/10km | 1 hourly |
| Reanalysis | ts2 | soil temperature | ° C | MERRA2 | 0.625°×0.5°/10km | 1 hourly |
| Reanalysis | ts3 | soil temperature | ° C | MERRA2 | 0.625°×0.5°/10km | 1 hourly |
| Remote Sensing | sm_s_wetness | surface soil wetness | unitless | SPL4SMGP.007 | 9 km | 3 hourly |
| Remote Sensing | sm_r_wetness | rootzone soil wetness | unitless | SPL4SMGP.007 | 9 km | 3 hourly |
| Remote Sensing | sm_p_wetness | profile soil wetness | unitless | SPL4SMGP.007 | 9 km | 3 hourly |
| Remote Sensing | nbar1 | red band | unitless | MCD43A4v061 | 500 m | daily |
| Remote Sensing | nbar2 | near infrared 1 band | unitless | MCD43A4v061 | 500 m | daily |
| Remote Sensing | nbar3 | blue | unitless | MCD43A4v061 | 500 m | daily |
| Remote Sensing | nbar4 | green | unitless | MCD43A4v061 | 500 m | daily |
| Remote Sensing | nbar5 | near infrared 2 band | unitless | MCD43A4v061 | 500 m | daily |
| Remote Sensing | nbar6 | shortwave infrared 1 band | unitless | MCD43A4v061 | 500m | daily |
| Remote Sensing | nbar7 | shortwave infrared 2 band | unitless | MCD43A4v061 | 500 m | daily |
| Remote Sensing | slope | terrain slope | radian | Geomorpho90m | 90 m | static |
| Remote Sensing | spi | stream power index | unitless | Geomorpho90m | 90 m | static |
| Remote Sensing | cti | compound topographic index | unitless | Geomorpho90m | 90 m | static |


## 2.3 Machine learning model

### 2.3.1 General model design

We used an RF regression algorithm to train site-level and grid-level ML models (Kim et al., 2020). RF regression builds an assembly of independent trees, each of which is trained from a random subset of input data and tested against the rest of the data (Breiman, 2001). A tree grows two leaves when a random selection of subset features reduces the mean squared error (MSE) of predictions after splitting at a leaf node. Each tree is trained on a bootstrap sample of input data. Trees constructed in this way are less correlated in the ensemble. The generalization error converges as the forest grows to a limit to avoid overfitting. Compared to other ML algorithms, RF has shown to have better accuracy and lower uncertainty (Irvin et al., 2021; Kim et al., 2020). This approach has been previously applied to upscaling $CH_4$ fluxes in wetlands and rice paddies across multiple ecosystems (Davidson et al., 2017; Feron et al., 2024; McNicol et al., 2023; Ouyang et al., 2023; Peltola et al., 2019).

A grid-search hyperparameter tuning for daily models was performed before predictor selection. We carried out analyses in Python version 3.6 with the ensemble RF regressor in package 'scikit-learn' (Pedregosa et al., 2011). With all the predictors and data, hyper-parameters were set after tuning for optimized model performance, including the number of trees (n_estimators=100), number of variables to consider when looking for the best split (max_features="sqrt", meaning the square root of the total number of feature variables), the maximum depth of the tree (max_depth=10), the minimum number of samples required to split a node (min_sample_split=10), and the minimum number of samples at a leaf node (min_samples_leaf=4).

For predictor selection and comparisons between the site-level model using *in situ* variables and the grid-level model using gridded versions of *in situ* variables, we built the model across all sites and adopted 5-fold cross-validation and 'out-of-bag' scores from ensemble trees to evaluate model performance, because, at this stage, we aimed to find physically reasonable variables from *in situ* measurements and to compare how the differences in scales and measuring methods between *in situ* predictors and gridded proxies affect model learned temporal variability in $CH_4$ fluxes. A subset of data was bagged to train each tree in the RF model, with the rest out-of-bag data used as independent validation data to evaluate the prediction accuracy of each tree, resulting in the average out-of-bag scores of all the trees in the model. Cross-validation was applied to daily predictions to select variables that can best predict the daily variability of $CH_4$ fluxes within sites. The changes in model performance after predictor selection and after switching from site-level variables (*in situ* measurements) to grid-level proxies (reanalysis data) were assessed, which helped quantify differences in model performance when modeling on *in-situ* measured predictor variables versus modeling on substitute variables at grid level. Because the data sources to model input from in situ versus from gridded variables were different, we separated site-level and grid-level modeling to ensure the importance of input features were comparable within a model. The feature importance reflects the relative importance of each input variable in a RF model. It also pertains to the input

data distribution and model structure. Therefore, the feature importance by site models can help
us identify controlling physical variables but would not necessarily translate to the same rank in
the feature importance of grid models, especially when additional gridded variables from remote
sensing products were added to complement the missing controllers from site models.
A summary of input variables for grid-level modeling is provided in Table 1. Although RF can
enhance model robustness when collinearity presents in input variables, the collinearity could
affect the interpretation of feature importance measured by impurity decrease in RF models.
Therefore, we first built a baseline grid-level model with independent variables after a pairwise
Pearson correlation test (Table S3) to exclude covariates. We grouped significantly correlated
variables ($p < 0.001$, $r > 0.8$, white grids except for those on the diagonal line in Table S3), forming
three groups: SMAP soil moisture variables in group 1 (we also included surface soil moisture
that was significantly correlated with the other two soil moisture variables and $r > 0.7$); air
temperature (tas), downward longwave radiation (rsdl), spfh, soil temperatures (ts1, ts2, and
ts3) in group 2; downward shortwave radiation (rsds) and latent heat (le) in group 3. We then
selected one most important variable in each group for the baseline models according to the
feature importance of modeling on all predictor variables (Fig. S14). The rest variables out of the
groups were included in the baseline features. The resulting baseline features included air
pressure (pa), latent heat flux (le), sensible heat flux (h), soil temperature (ts2), rootzone soil
wetness (sm_r_wetness), slope, spi, and cti. Then we designed four additional different model
settings by changing predictor variables, including (1) baseline variables plus covariates, (2)
only variables from MODIS NBAR, (3) baseline variables plus NBAR bands, and (4) all predictor
variables. In this forward feature selection process, we evaluated the impacts of adding
constraint variables from remote sensing products on model performance.
Model predictive performance evaluates the accuracy of a model to predict at a new site without
any prior knowledge. For the spatial predictive performance evaluation of grid-level ML models,
we used a nested leave-one-site-out cross-validation scheme (LOOCV, hereafter). Such a
scheme selects one site to use as independent validation data to evaluate models trained and
tested with data from the remaining sites, repeating the process for all sites. Without any prior
knowledge of the validation site added to a model, the LOOCV scheme can assess the
predictive ability of the model in a new place as well as evaluate the uniqueness of a site in the
dataset. Similar forms of spatial LOOCV have been used to evaluate upscaling models for
global or regional $CO_2$ and $CH_4$ (McNicol et al., 2023; Peltola et al., 2019; Virkkala et al., 2021).
The validation of the upscaling model was not only performed with respect to daily predictions,
but also on monthly means. The predictive performance of the upscaling model on monthly
variability of $CH_4$ fluxes and spatial variability across sites is important for studies that vary in
temporal and spatial scales.
Model predictive performance was assessed using three evaluation metrics: mean absolute
error (MAE), root mean squared error (RMSE), and $R^2$ score. Daily modeled $CH_4$ fluxes were
compared to EC observations at each validation site. The evaluation metrics were calculated at
daily and monthly scales for each site separately to examine the model performance by general
wetland types and for all sites pooled together to evaluate the overall performance and compare
with existing studies. Squared error metrics are more sensitive to outliers and highly skewed
data, which is often the case with $CH_4$ fluxes. Therefore, we selected both MAE and RMSE to
quantify the errors. The mean error (ME) between model predictions and validation data was
calculated, representing systematic bias in predicted fluxes. The standard deviation of model
residuals was also included to measure the spread of the residuals. This matches RMSE when
ME equals zero.
Two additional ML algorithms were compared with RF: SVM and ANN. SVM is efficient with
sparse data where the dimension of the input space is greater than the number of training
samples (Kuter, 2021). While the training process of ANN is expensive and time-consuming, it
can develop deep networks with growing training data which may increase predictive
performance (Saikia et al., 2020). We used support vector regression to model $CH_4$ fluxes with
the same predictor variables and dataset as used in ensemble RF regressions. Multilayer
perceptron regressor is an implementation of an ANN model that adjusts the weights of neurons
using backpropagation to improve prediction accuracy. It uses the square error as the loss
function and a stochastic gradient-based optimizer 'adam' for weight optimization. We used two
hidden layers in the ANN model, each with 50 neurons. Data from all variables were normalized
to achieve the best model performance of SVM and ANN.

## 2.3.2 $CH_4$ flux upscaling


We trained 500 ensemble RF models with all gridded predictors of grid-level models from the
general model design and with data from all sites for upscaling daily $CH_4$ fluxes. Each RF model
was trained with the same optimized hyper-parameters and different bootstrap samples.
Ensemble models were then applied to 0.098° gridded predictors to produce the upscaling $CH_4$
flux intensities from the means of the 500 predictions and the prediction uncertainty from the
standard deviations. Given that the $CH_4$ fluxes were modeled with data from the wetland EC
sites, a wetland extent map was also needed to constrain the areas when scaling grid emissions
(see section 2.4). Final $CH_4$ emission and uncertainty maps associated with wetland extents
were the results of multiplying the predicted means and standard deviations of flux intensities
with wetland areas. All wetland maps were resampled to 0.098° x 0.098° resolution with a
conservative remapping method for producing the emission products.

# 2.4 Wetland extent maps and benchmark estimates of wetland $CH_4$ emissions



Wetland extent maps were applied to scale the modeled $CH_4$ flux intensities to the region. The
Wetland Area and Dynamics for $CH_4$ Modeling (WAD2Mv2), representing spatiotemporal
patterns of inundated vegetated wetlands at 0.25° resolution, was selected as the reference for
dynamic wetland areas in this study (Zhang et al., 2021). Active and passive microwave
detected inundation combined with static wetlands were used to delineate the monthly dynamics
of wetland inundation between 2000 and 2020. Open water bodies such as lakes, rivers,
reservoirs, coastal wetlands, and rice paddies were excluded. We used monthly mean WAD2M
fractions between 2010 and 2020 to represent seasonal wetland dynamics. Emission
estimations are subject to differences in the wetland extent between maps (Saunois et al.,
2020). We used monthly means of the Global Inundation Extent from Multi-Satellites (GIEMS2)
product (Prigent et al., 2020) to represent temporal patterns of the restricted wetland extents at
0.25° resolution. The coarse resolution maps were resampled to 0.098° x 0.098° grids using the
nearest neighbor method. The static Global Lakes and Wetlands Database version 1 (GLWDv1)
Level 3 1-km resolution map excluding classes of lakes, rivers, and reservoirs (Lehner and Döll,
2004) was included to quantify the upper limit of wetland cover. For all explicit GLWDv1 wetland
classes, we assumed a 100% wetland coverage in the original pixels, except for 'intermittent
wetland/lake' for which we assumed a 50% coverage; for GLWDv1 classes represented as
extent ranges, we used the average value of the range (i.e., 75% for 50-100% wetland, 37% for
25-50% wetland, and 12% for 0-25% wetland). To support domain emission comparisons,
wetland cover was also extracted from the updated GLWD version 2 dataset (GLWDv2, Lehner
et al., 2024) which provides the spatial extent of 33 waterbody and wetland classes at 500-m
spatial resolution. All freshwater wetland classes that occur in our study area (classes 8-25)
from GLWDv2 were included (i.e., excluding rivers, lakes, reservoirs and other permanent open
water bodies, as well as coastal saline/brackish wetlands). The original wetland areas per
GLWDv2 pixel were summed across all included classes to derive a total wetland area per pixel.
Furthermore, a regional freshwater wetland distribution dataset was calculated from a
permafrost region specific land cover map (CALU - circum-Arctic landcover units) which
classified 23 land covers including 3 wetland classes and 10 moist to wet tundra classes at 10-
m resolution and aggregated to 1km with the majority class (Bartsch et al., 2024). This regional
wetland map was applied for $CH_4$ emission estimation in the North Slope region in Alaska to
assess the impacts of different wetland maps on emission estimates in this area when
compared against airborne measurements. Wetland areas from the finer resolution maps were
aggregated to 0.098° x 0.098° grids for emission calculations.
We compared WetCH$_4$ emissions with benchmark domain or regional estimates from bottom-up
process models, top-down atmospheric observation-based inversions, and existing upscaling
studies. We acquired data for the study domain from the ensemble mean of bottom-up process-
based models from the Global Carbon Project (GCP) (Zhang et al., 2025) and the extended
ensemble of wetland $CH_4$ estimates that were priors for the top-down GEOS-Chem atmospheric
chemical and transport model (WetCHARTs) (Bloom et al., 2017; Friedlingstein et al., 2022).
We also included the atmospheric inversions of northern high latitudes from an assimilation
CarbonTracker-$CH_4$ system (Bruhwiler et al., 2014 update at
https://gml.noaa.gov/ccgg/carbontracker-ch4/carbontracker-ch4-2023/). We compared WetCH$_4$
with existing upscaled products of monthly $CH_4$ wetland fluxes based on Peltola et al. (2019) for
the study domain. For regional wetland hotspots, $CH_4$ flux estimates were obtained from Carbon
in Arctic Reservoirs Vulnerability Experiment (CARVE), which measured total atmospheric
columns of $CO_2$, $CH_4$, and carbon monoxide over North Alaska in spring, summer, and early fall
between 2012 and 2014 (Chang et al., 2014; Miller et al., 2016). These were used to verify our
seasonal emission estimates over the North Slope region (Zona et al., 2016).

# 516  3. Results

## 517  3.1 Model validation

### 518  3.1.1 Site-level modeling

Site-level modeling used all wetland sites to build a RF model and identified the 10 most
important variables measured *in situ* that, if left out, decreased the valuation score of the model
by more than 90% based on the mean decrease in impurity (Fig. S3). With bootstrap sampling
and using all candidate predictors (Fig. 1) in the model, the out-of-bag RMSE of the site-level
model was 30.22 nmol m$^{-2}$ s$^{-1}$, and the out-of-bag $R^2$ between observed daily means of $CH_4$
fluxes and prediction was 0.73. Modeling with the 10 most important variables at site level
resulted in similar model performance, with an out-of-bag RMSE of 30.43 nmol m$^{-2}$ s$^{-1}$ and an
out-of-bag $R^2$ of 0.73. Site-level model performance converged as the increment of predictor
variables ordered by the importance rank (Fig. S4). We then tested building separate models
according to wetland types because distinct $CH_4$ fluxes have been observed from wet tundra
(Fig. S5, mean ± standard deviation: 13 ±14 nmol m$^{-2}$ s$^{-1}$), bogs (22 ±26 nmol m$^{-2}$ s$^{-1}$) and fens
(56 ±88 nmol m$^{-2}$ s$^{-1}$). The out-of-bag $R^2$ (RMSE) was 0.85 (7.2 nmol m$^{-2}$ s$^{-1}$) for bog, 0.84 (27.7
nmol m$^{-2}$ s$^{-1}$) for fen, and 0.57 (34.3 nmol m$^{-2}$ s$^{-1}$) for wet tundra. Modeling with the selected 10
predictors resulted in an out-of-bag $R^2$ (RMSE) of 0.84 (7.6 nmol m$^{-2}$ s$^{-1}$) for bog, 0.84 (27.9
nmol m$^{-2}$ s$^{-1}$) for fen, and for 0.53 (36.3 nmol m$^{-2}$ s$^{-1}$) wet tundra. Next, we tested whether the
inclusion of non-wetland sites (upland and rice sites) would affect model performance. This
resulted in an out-of-bag $R^2$ decrease to 0.56 and RMSE increase to 38.86 nmol m$^{-2}$ s$^{-1}$, which
suggests that a generalized ML model over all land cover classes is not practical to reliably
predict $CH_4$ fluxes with the current set of predictors and available data. This is most likely due to
the distinctive features of $CH_4$ emissions between wetlands and non-wetland classes (Fig. S5).

### 539  3.1.2 Grid-level modeling and remote sensing constraints

Substituting *in-situ* measurements of selected predictor variables with gridded MERRA2
variables slightly reduced model accuracy. The out-of-bag $R^2$ decreased by 9.6% to 0.65 and
RMSE increased by 15% to 34.9 nmol m$^{-2}$ s$^{-1}$ compared to the site-level model. The coarse
resolution MERRA2 reanalysis data captures less spatial variability of the selected physical
variables and is less accurate at the grid-level compared to *in situ* EC measurements.
Adding remote sensing constraints to the gridded variables can improve model predictive
performance and reduce errors. Modeling on baseline features explained on average 46% of
daily $CH_4$ fluxes' variability in validation sites with the largest range of errors (Fig. 3a). The
medians in the baseline model of $R^2$, MAE, RMSE, ME under the LOOCV scheme were 0.5,
16.4 nmol m$^{-2}$ s$^{-1}$, 21.0 nmol m$^{-2}$ s$^{-1}$ and 6.4 nmol m$^{-2}$ s$^{-1}$, respectively. Adding NBAR or
covariates from MERRA2 and SMAP input variables returned a higher mean $R^2$ or slightly lower
mean errors than the baseline model, whereas modeling with all gridded input variables (the 'all'
model setting) achieved the highest mean $R^2$ of 0.51 with the comparable mean MAE (23.6
nmol m$^{-2}$ s$^{-1}$), RMSE (32.1 nmol m$^{-2}$ s$^{-1}$) and ME (0.9 nmol m$^{-2}$ s$^{-1}$) (Table S4). Although
modeling with baseline features and covariates (the 'base+CoVar' setting) received a
comparable mean $R^2$ with modeling 'all' variables, the latter had a higher median $R^2$ (0.53) and
lower median errors (MAE=14.1nmol m$^{-2}$ s$^{-1}$, RMSE=19.8 nmol m$^{-2}$ s$^{-1}$, ME=4.0 nmol m$^{-2}$ s-1).
Our results suggest that including remote sensing constraints or covariates improved models'
ability to predict spatial variability in wetland $CH_4$ fluxes and reduced prediction errors. These
results confirm our selection of predictor variables for the upscaling model (Table 1).
The average importance of the baseline features shows their influence on the grid-level model
predictive performance (Fig. 3b). Importance of independent predictors under LOOCV scheme,
though slightly varied between models, agreed in selecting MERRA2 soil temperature (ts2) as
the primary driver in predicting daily $CH_4$ fluxes in northern wetlands, followed by SMAP
rootzone wetness (sm_r_wetness). The eight baseline features accounted for a 99% reduction
in the mean validation score of the baseline models. The average importance of 'all' gridded
variables used for upscaling (Fig. S14) was consistent with baseline models, emphasizing the
importance of soil temperatures and rootzone wetness. Additionally, air pressure and
topography also contributed to explaining the daily variability in $CH_4$ fluxes. Nevertheless, all
variables contributed to predicting variability in $CH_4$ fluxes, showing the complexity of
environmental factors that would affect the rates of $CH_4$ production and the process of gas
exchange.

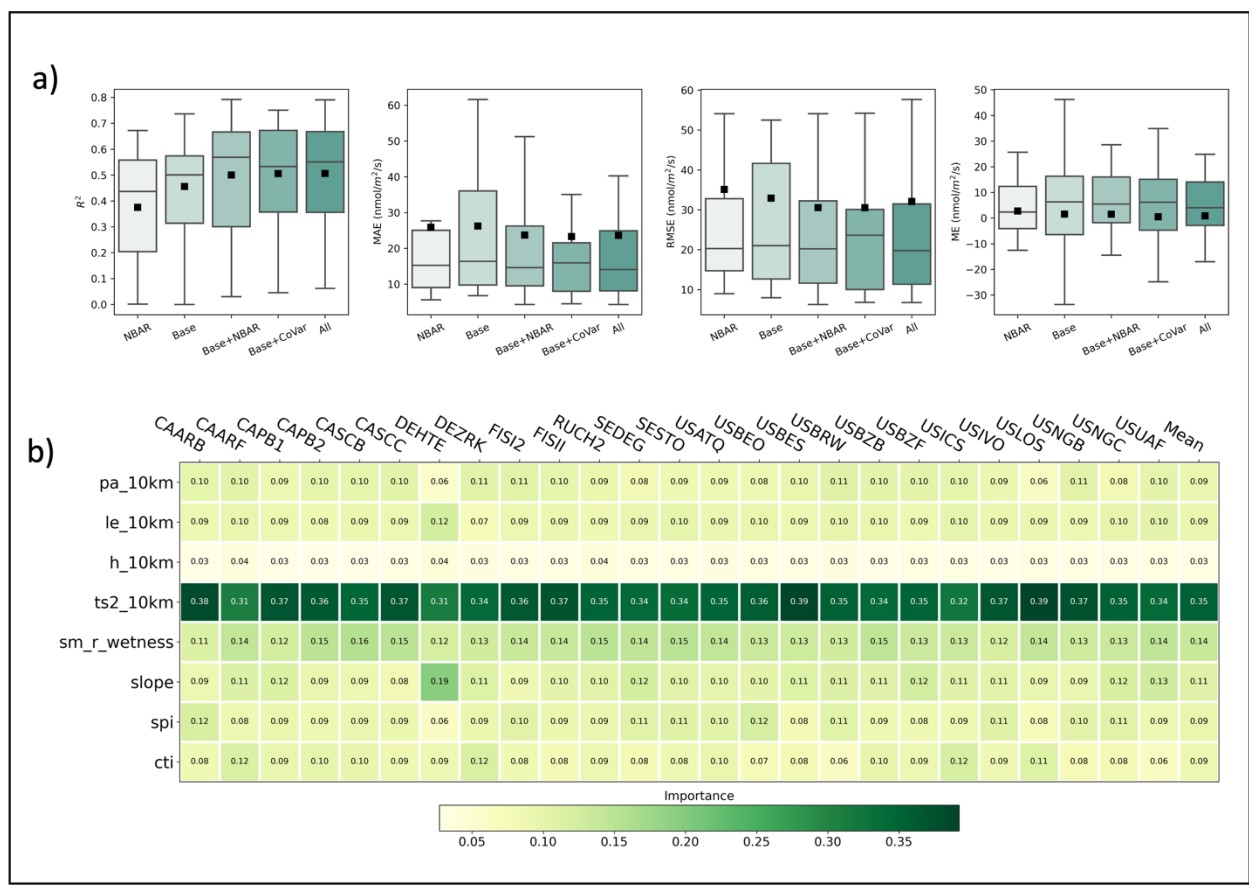

Figure 3 Grid-level modeling: a) Distribution of $R^2$, MAE, RMSE, and ME for all sites (size = 25)
in a LOOCV scheme based on gridded data using five model settings: RF modeled using only
MODIS NBAR bands, baseline features (MERRA2 air pressure, latent heat flux, sensible heat
flux, soil temperature, SMAP rootzone soil wetness, topographic slope, spi, and cit), baseline
features plus MODIS NBAR bands, baseline features plus correlated variables within the
MERRA2 and SMAP dataset, and all gridded input variables together. The model settings are
ranked by mean $R^2$, from lowest (left) to highest (right); b) Mean variable importance of baseline
models (last column) in the LOOCV scheme and at each site (columns labeled with validation
site ID). The values in each column are the means of accumulation of the impurity decrease
when a variable was taken out in the trees of a RF model, representing the importance of such
variable to the model. The variable names and descriptions refer to Table 1.
Daily mean $CH_4$ fluxes exhibited great variability in wetlands across space and time (mean = 35
nmol m$^{-2}$ s$^{-1}$, σ = 65 nmol m$^{-2}$ s$^{-1}$, Fig. S3). The model predictive performance (Fig. 4) was
calculated for each site and the average performance on the daily variability in $CH_4$ fluxes was
best at wet tundra sites with a mean $R^2$ of 0.56, followed by bog sites (0.51) and fen sites (0.45).
Due to the large variability in fen daily fluxes, errors of daily predictions were highest in fen sites
(mean RMSE = 54.2 nmol m$^{-2}$ s$^{-1}$ and mean MAE = 37.8 nmol m$^{-2}$ s$^{-1}$), followed by bog sites
(mean RMSE = 27.6 nmol m$^{-2}$ s$^{-1}$ and mean MAE =22.5 nmol m$^{-2}$ s$^{-1}$),  and were lowest in wet
tundra sites (mean RMSE = 13.5 nmol m$^{-2}$ s$^{-1}$ and mean MAE =10.3 nmol m$^{-2}$ s$^{-1}$). Our model
slightly overestimated daily fluxes (mean ME = 0.9 nmol m$^{-2}$ s$^{-1}$) was driven by underestimation
of fen sites (mean ME = -12 nmol m$^{-2}$ s$^{-1}$) versus overestimation of bog (mean ME = 14 nmol m$^{-2}$
s$^{-1}$) and wet tundra (mean ME = 3 nmol m$^{-2}$ s$^{-1}$) sites.
Model predictive performance on aggregated monthly means of CH$_4$ fluxes increased by 37%
as compared to daily means (mean R$^2$ = 0.70, Fig.4, Table S4). This improvement may be
attributed to a better representation of the environmental conditions' average state over a month
by the input variables compared to the daily variability. Performance was higher in wet tundra
(mean R$^2$ = 0.73) and bogs (mean R$^2$ = 0.73) and lower in fen sites (mean R$^2$ = 0.64, Fig. 4).
Mean errors in monthly mean predictions were: RMSE = 28.1 nmol m$^{-2}$ s$^{-1}$, MAE = 21.4 nmol m$^{-2}$
s$^{-1}$, and ME = 0.37 nmol m$^{-2}$ s$^{-1}$ (Table S4). Prediction residuals of daily and monthly CH$_4$ fluxes
(Fig. S6) showed normal distributions for wet tundra sites, indicating the spread of residuals
were random errors that increased with the flux magnitude. The residuals had a skewed normal
distribution for bog sites indicating likely overestimation. The long-left tails in prediction residuals
indicated that the intense emission fluxes from fens during summer peaks were underestimated
(Fig. S6).
Site-by-site validation of daily flux predictions varied greatly between individual sites (Fig. 5, S7).
For example, US-UAF, an EC site in Interior Alaska with mature black spruce cover and full
understory vegetation and mosses over permafrost (Ueyama et al., 2023a), which is the only
one of the five forest bog sites in our dataset that had low CH$_4$ fluxes and weak seasonal cycles
(less than 10 nmol m$^{-2}$ s$^{-1}$), was significantly overestimated by our model (RMSE = 58 nmol m$^{-2}$
s$^{-1}$ and MAE = 53 nmol m$^{-2}$ s$^{-1}$). Permafrost presence and ground water below soil surface may
explain the low fluxes at this site (Iwata et al., 2015; Ueyama et al., 2023b).

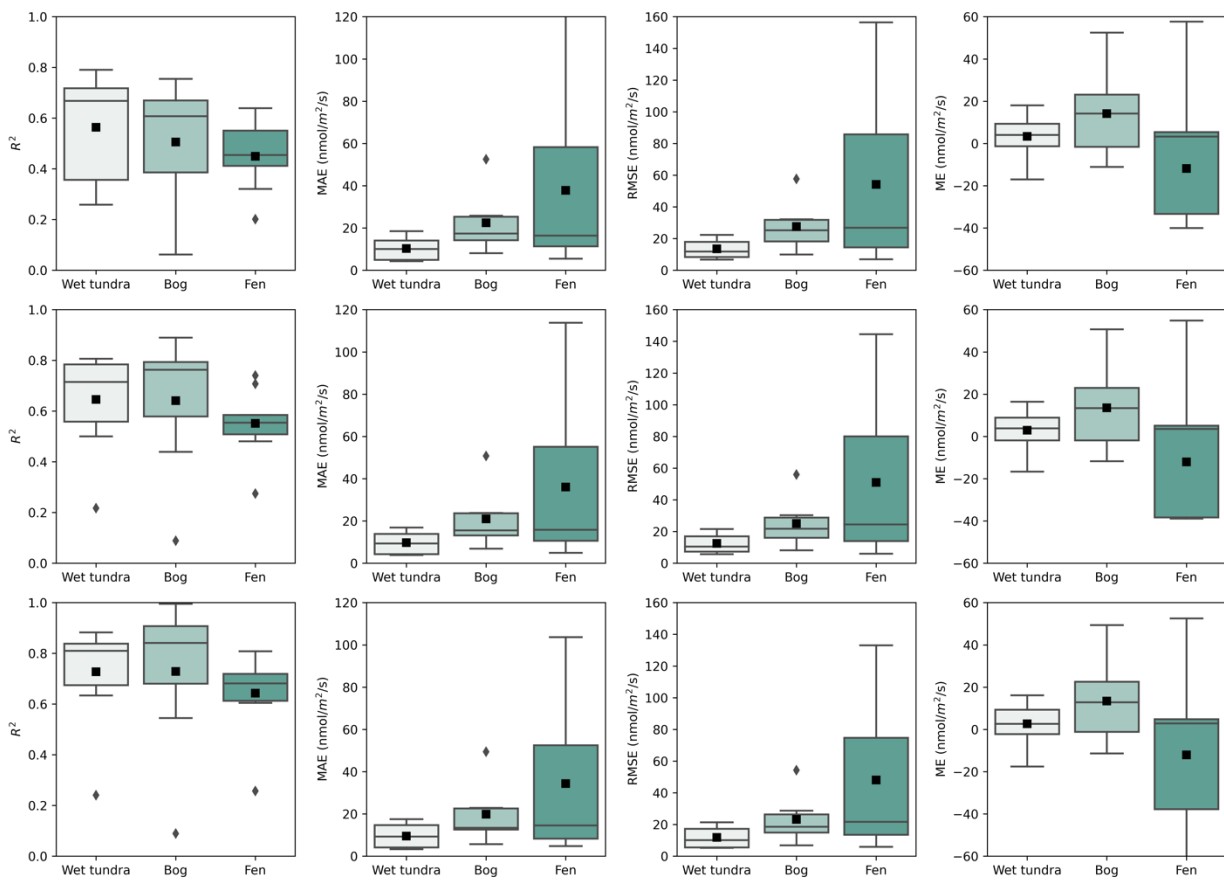

Figure 4 Model predictive performance evaluation on RF modeled $CH_4$ fluxes at grid level under
LOOCV scheme: boxplots of $R^2$, MAE, RMSE, and ME across validation sites by wetland types
with mean values denoted in black squares at daily/weekly/monthly (top/middle/bottom panel)
time steps.

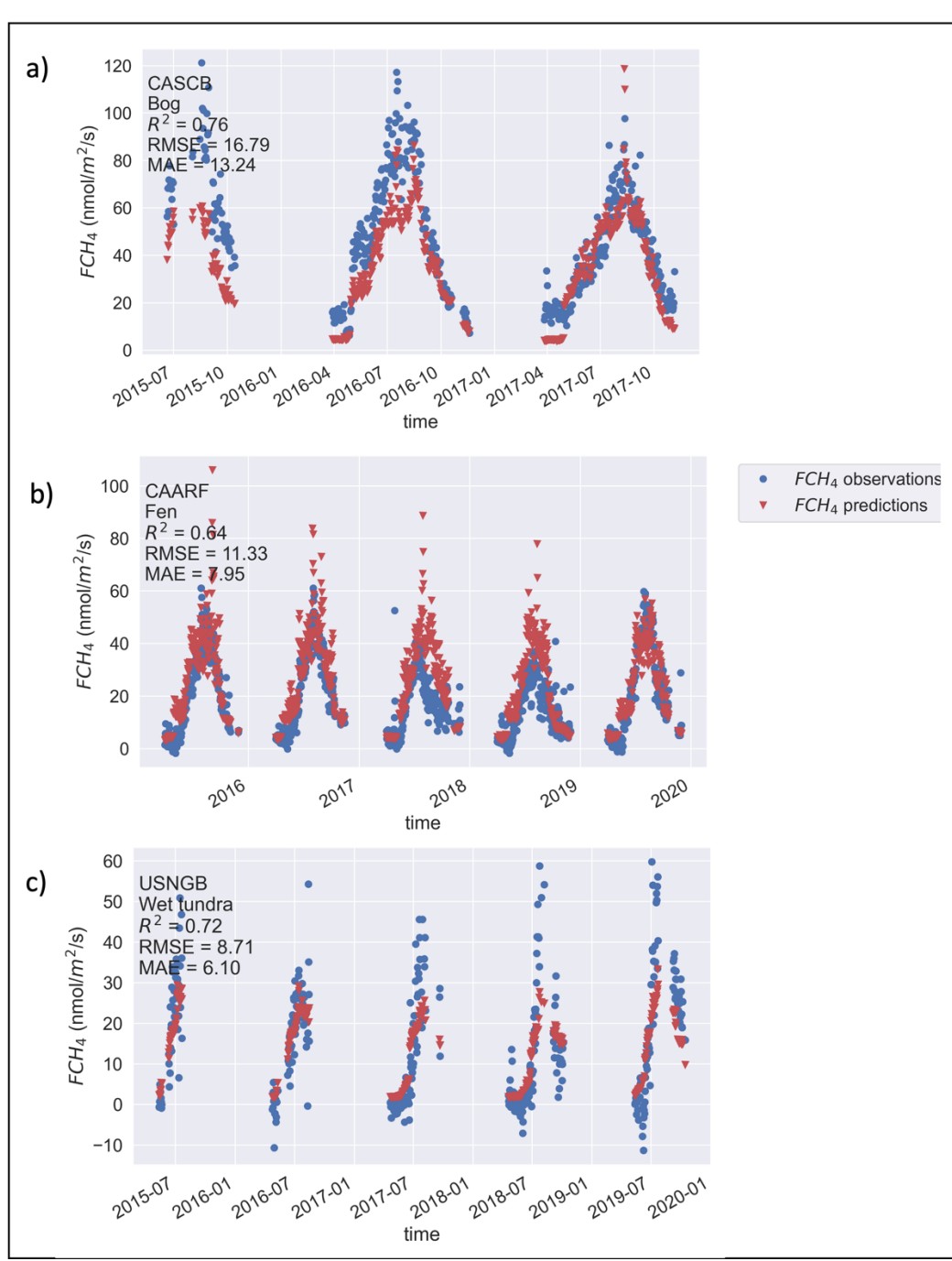

Figure 5 Example model predictive performance in seasonal cycles of daily FCH$_4$ at the
validation sites of CA-SCB, CA-ARF, and US-NGB, representing bog, fen, and wet tundra,
respectively.

## 3.2 Upscaled wetland $CH_4$ emissions

### 3.2.1 Wetland area weighted $CH_4$ emissions

Upscaled daily $CH_4$ fluxes were weighted by wetland fraction to estimate gridded daily $CH_4$ fluxes from northern wetlands based on WAD2Mv2, GIEMS2, and GLWDv1 between 2016 and 2022 (Fig. 6), and GLWDv2 for comparison. The mean annual emissions and RF model associated uncertainties are summarized with different wetland maps in Table S5. The estimate from WetCH4 with WAD2Mv2 was 22.8 ±2.4 Tg $CH_4$ $yr^{-1}$, comparable to UpCH4 (23.5 ±5.8 Tg $CH_4$ $yr^{-1}$). With GIEMS2, WetCH4 estimated the minimum annual emission of 15.7 ±1.8 Tg $CH_4$ $yr^{-1}$. With GLWDv1 and GLWDv2, WetCH4 estimated potential annual emissions of 46.0 ±5.1 Tg $CH_4$ $yr^{-1}$ and 51.6 ±2.2 Tg $CH_4$ $yr^{-1}$ for 2016-2022, respectively. The spatial patterns were similar to the post 2016 mean annual fluxes from the GCP process-model ensemble means (28.6 ±21.6 Tg $CH_4$ $yr^{-1}$ for 2016-2020), WetCHARTs (29.5 ±30.0 Tg $CH_4$ $yr^{-1}$ for 2016-2019), and atmospheric inversions of CarbonTracker-CH4 (40.9 Tg $CH_4$ $yr^{-1}$ for 2016-2022), highlighting the high emission areas in the Hudson Bay Lowlands and West Siberian Lowlands. The emissions from WetCH4-GIEMS2 were lower in these two hotspots than other estimates. Differences in the distribution of $CH_4$ emissions between wetland products reflect the influence of wetland dynamics. Mean monthly wetland inundations are provided by WAD2Mv2 and GIEMS2, which set the dynamic limits for the wetland boundaries of the $CH_4$-emitting surface. While emissions resulting from inundation were captured, it appeared that saturated or wet subsoil conditions were not well represented by WAD2M and GIEMS2, resulting in low emissions in wet yet non-inundated tundra (i.e., Alaska North Slope). To address this, we incorporated wetland fractions from the CALU high-resolution wetland map (Bartsch et al., 2024) specifically produced for the permafrost region in order to estimate Alaska North Slope emissions. Wetland fractions from GLWD (both v1 and v2) represent a static maximum wetland distribution throughout time. Thus, estimates from GLWD may represent the upper bounds for all northern wetlands under contemporary conditions.

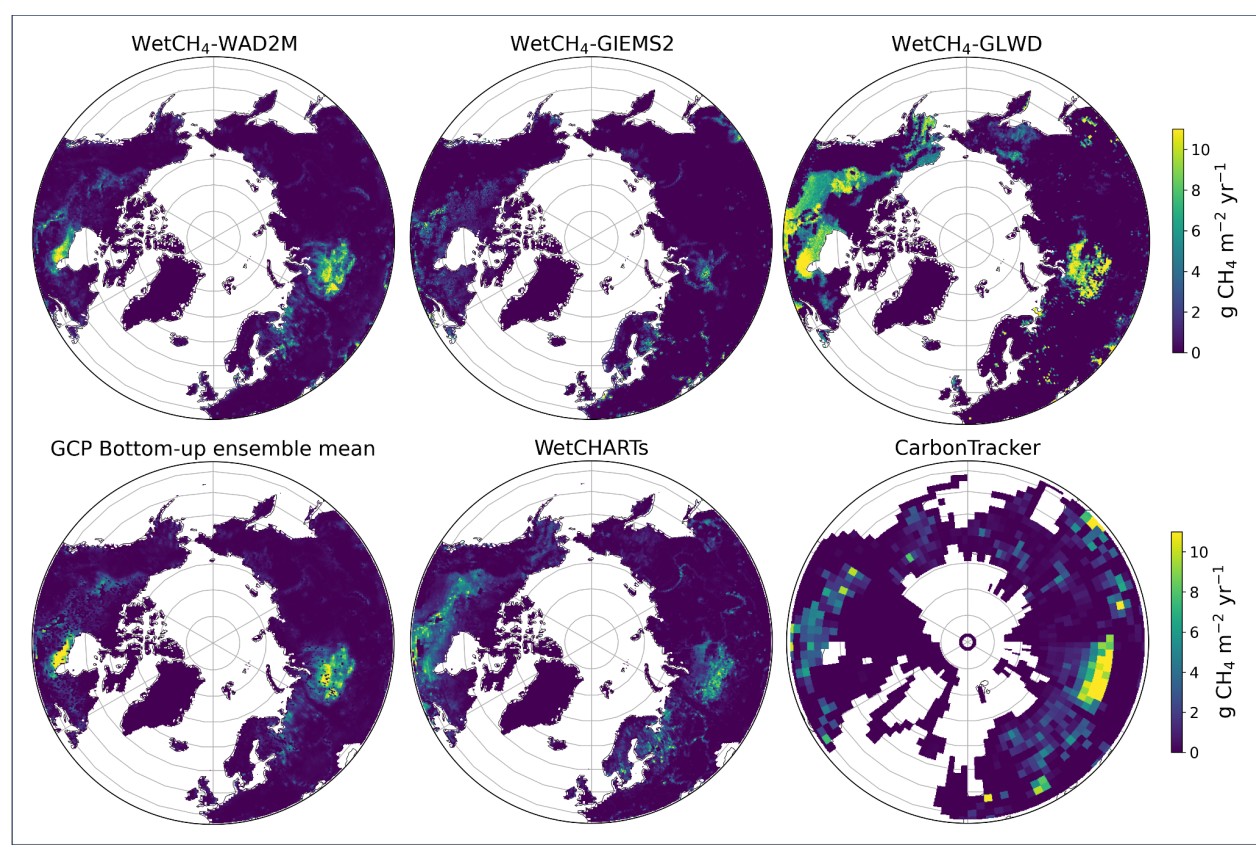

Figure 6 Mean annual wetland $CH_4$ fluxes: the top row contains $WetCH_4$ upscaled fluxes
between 2016 and 2022 and weighted by wetland fractions for three wetland maps WAD2Mv2,
GIEMS2, and GLWDv1; the bottom row contains bottom-up GCP ensemble mean,
WetCHARTs, and top-down estimates of CarbonTracker-$CH_4$ natural microbial emissions.
We compared spatial distributions of our upscaled fluxes ($WetCH_4$) with two alternative
upscaled datasets. Using the same wetland weights, our product showed similar spatial patterns
to $UpCH_4$ (McNicol et al., 2023) and the upscaled fluxes from Peltola et. al. (2019) (Fig. S9).
Spatially, the maximum mean flux of 2016-2022 for $WetCH_4$ with WAD2Mv2 was 69 mg $CH_4$ $m^{-2}$
$day^{-1}$, $UpCH_4$ produced a maximum mean flux between 2016-2018 of 88 mg $CH_4$ $m^{-2}$ $day^{-1}$.
While all three products predicted concentrated $CH_4$ exchange in the Hudson Bay Lowlands and
West Siberian Lowlands, and low fluxes in West Canadian Arctic tundra, $WetCH_4$ predicted
lower fluxes in forested wetlands of West Canada than $UpCH_4$ (Fig. S9 a,b). With GLWDv1,
$WetCH_4$ predicted similar fluxes to those of Peltola et al. (2019), with the exception of a number
of potent emitting grids in the West Siberian Lowlands (Fig. S9 c,d) and a maximum mean flux
of 132 mg $CH_4$ $m^{-2}$ $day^{-1}$ from $WetCH_4$.
## 3.2.2 Seasonal cycles of wetland $CH_4$ emissions
Mean seasonal cycles of wetland $CH_4$ emissions were consistent with bottom-up estimates in
the domain and top-down inversions in high latitudes (Fig. 7). The amplitudes of two ML-based
estimates agreed in the domain ($WetCH_4$ and $UpCH_4$ both within WAD2Mv2 wetland areas) and
were lower than the ensemble means of GCP or WetCHARTs estimates during the growing
season (Fig. 7a). In the northern high latitudes (60° - 90° N), the amplitudes of this study closely
agree with WetCHARTs, and both were lower than the ensemble means of GCP in the growing
season (Fig. 7b). Our emissions in June-July-August were lower than the emissions attributed
by the atmospheric inversion of CarbonTracker-$CH_4$, which does not discriminate between
wetland and open water sources. We did not use comparisons with CarbonTracker-$CH_4$ for 45°-
90° due to likely considerable contributions from aquatic systems and other non-wetland factors
in the inversion estimates. Notably, uncertainties between ML-based approaches with the same
wetland extents showed less variation than those between process-based models, especially
during the growing season. The phase of our estimates (WetCH$_4$) agreed with bottom-up and
top-down models, peaking in July followed by August (Fig. 7a,b), whereas UpCH$_4$ showed a
month lag, probably due to the two- or three-week lag of predictor variables selected in UpCH$_4$
(McNicol et al., 2023). Peak fluxes in July and August were commonly seen in tower
measurements.
The seasonality in upscaled wetland $CH_4$ emissions corresponded to the intensities of fluxes
and dynamics of wetland areas. We compared mean seasonal cycles of upscaled products with
different dynamic or static wetland maps to constrain the impacts of wetland areas (Fig. 7c). As
observed in spatial distributions (Fig. 7a,c), emissions from the potential emitting surface
(WetCH$_4$_GLWDv1) were 95% higher than those from reference inundated wetlands
(WetCH$_4$_WAD2Mv2) during the growing season, and doubling in winter. Within the GLWDv1
emitting surface, WetCH$_4$ predicted higher emissions than Peltola et al. (2019) in July (43%),
August (43%), December (41%), and January (61%), but 15% lower in October. We decoupled
the mean annual seasonal cycle for WAD2M from the emission seasonality by using a fixed
maximum WAD2M extent. The addition of maximum annual wetland extent further constrains
the limitations of seasonal WAD2M extents in underestimating methane emitting surface for
northern high latitude wetlands, especially in cold seasons. The resulting seasonal emissions
primarily driven by soil temperatures and moisture manifested elevated emissions in all months
and an intensified seasonal cycle. Reported emissions (Zona et al., 2016) and large bursts
(Mastepanov et al., 2008) from the freezing active layer at permafrost areas in October (zero-
curtain period) may not be well captured by our ML model. The differences in wetland areas
between the two dynamic products (WAD2Mv2 and GIEMS2) mostly affected emissions in May
and June in WetCH$_4$, but significantly affected emission magnitudes in UpCH$_4$. Despite the
differences in wetland areas, the phases of emissions cycles of WetCH$_4$ were consistent with
those from Peltola et al. (2019), whereas UpCH$_4$ again lagged a month.
We compared upscaled seasonal cycles with $CH_4$ fluxes estimated from regional airborne
measurements taken during CARVE campaigns over the Alaska North Slope (Fig. 7d). Given
that the wetland area in this region is uncertain (Miller et al., 2016), we computed mean
seasonal cycles over the land assuming all land in this area is water saturated in the soil, over
freshwater wetlands of CALU, and over WAD2M and Hydrolakes, representing three different
scenarios. In the lowland area of the North Slope (74295 km$^2$ spanning between 69.8°N -
71.4°N, 164.4°W - 152.7°W), the wetland area was estimated at 10611 km$^2$ from CALU, 4800
km$^2$ from GLWDv2, and 4049 km$^2$ from the maximum extent month in July of WAD2Mv2,
respectively. The range of our upscaled estimates aligned with regional emissions derived from
CARVE measurements. Chang et al. (2014) estimated 7 ±2 mg $CH_4$ m$^{-2}$ d$^{-1}$ of mean $CH_4$ fluxes
during the growing season in the North Slope from the column analysis of CARVE data. The
mean fluxes (May to September) of Wet$CH_4$ with CALU were estimated at 7.3 ±0.8 mg $CH_4$ m$^{-2}$
d$^{-1}$ (5.5 ±0.6 mgC $CH_4$ m$^{-2}$ d$^{-1}$), which is within the range of various CARVE estimations (Miller et
al., 2016). The landscape is in the biome of the Arctic coastal tundra and is covered by sedges,
grasses, mosses, and dwarf shrubs. A large number of lakes and freshwater ponds are
scattered across the area. Studies at the West Alaska lowland of Yukon–Kuskokwim Delta
found aquatic fluxes that were about ten times higher than in wet tundra during September
(Ludwig et al., 2023), suggesting that a major source of the airborne fluxes missing in Wet$CH_4$ in
the late growing season, can be attributed to open water fluxes. Remarkable increases could be
in summer and winter if we assume wetland over this region, as indicated by the range between
the green and the black lines in Fig. 8d. Yet, future emissions due to permafrost thaw still
depend on the hydrological changes of the landscape.

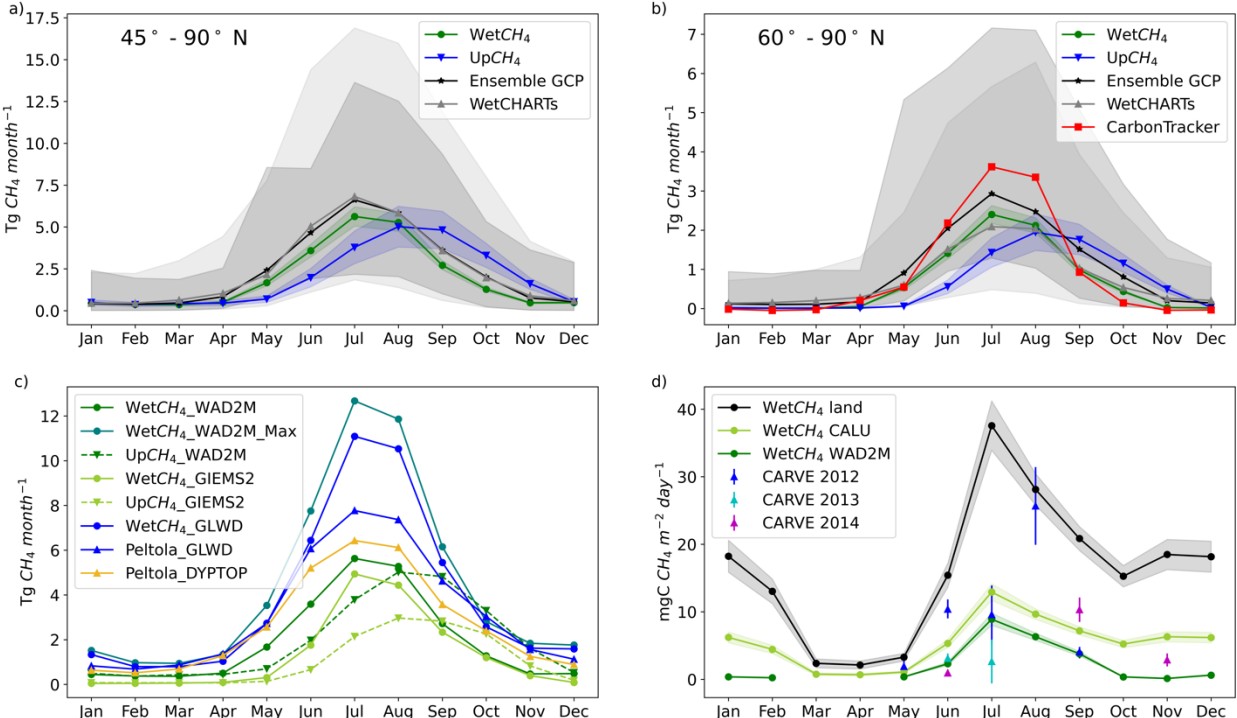

Figure 7 Multi-year average seasonal cycles of wetland $CH_4$ emissions: (a) comparison of ML
upscaled mean seasonal cycles in reference wetland areas (WAD2Mv2) with the cycles from
process-based models in the northern mid-high latitudes (45° - 90° N); (b) same comparison for
northern high latitudes (60° - 90° N) and addition of atmospheric CarbonTracker-$CH_4$ attributed
microbial emissions (2016-2022); (c) comparison of three ML upscaled mean seasonal cycles of
$CH_4$ emissions with different wetland area maps (WAD2Mv2, WAD2Mv2 maximum extent,
GIEMS2, GLWDv1); (d) comparison of Wet$CH_4$ mean seasonal cycles over the land (black line),
weighted by wetland of the CALU map (olive line), or weighted by fractions of WAD2Mv2 (green
line), with estimates of $CH_4$ fluxes in growing seasons from CARVE retrievals in North Slope
area of Alaska (Zona et al., 2016).

### 3.2.3 Interannual variations in wetland $CH_4$ emissions


The mean annual emissions from ML-based estimates with WAD2M were lower than the GCP bottom-up ensemble mean and WetCHARTs over different years from 2016 forward (Fig. 8a). All products demonstrated similar emission patterns for the domain in the interannual trends and variations, highest in 2016 and lower for three years from 2017 to 2019 (Fig. 8). The interannual variations in WetCH$_4$ were driven by the interannual variability in the upscaled fluxes as only multi-year mean seasonal dynamics from WAD2Mv2 were used. All products identified intensified emissions in 2016 as indicated by the variations relative to period means (Fig. 8b). Higher than period average emissions in 2020 were also modeled by WetCH$_4$ and ensemble

GCP.

a)

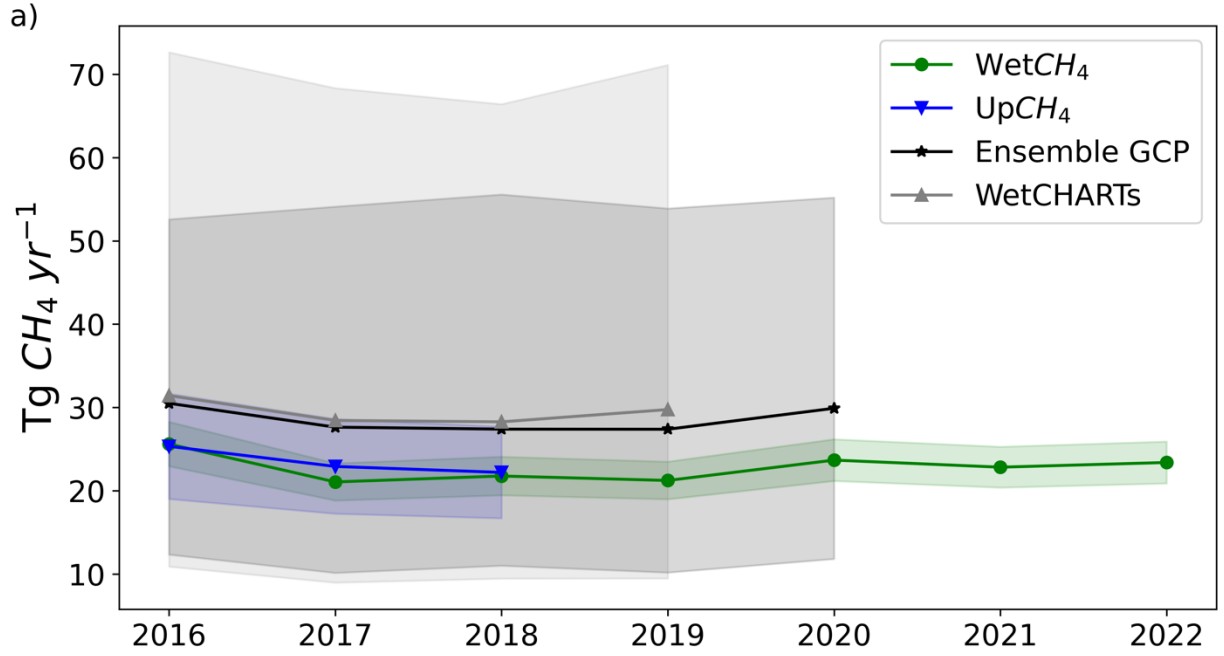

b)

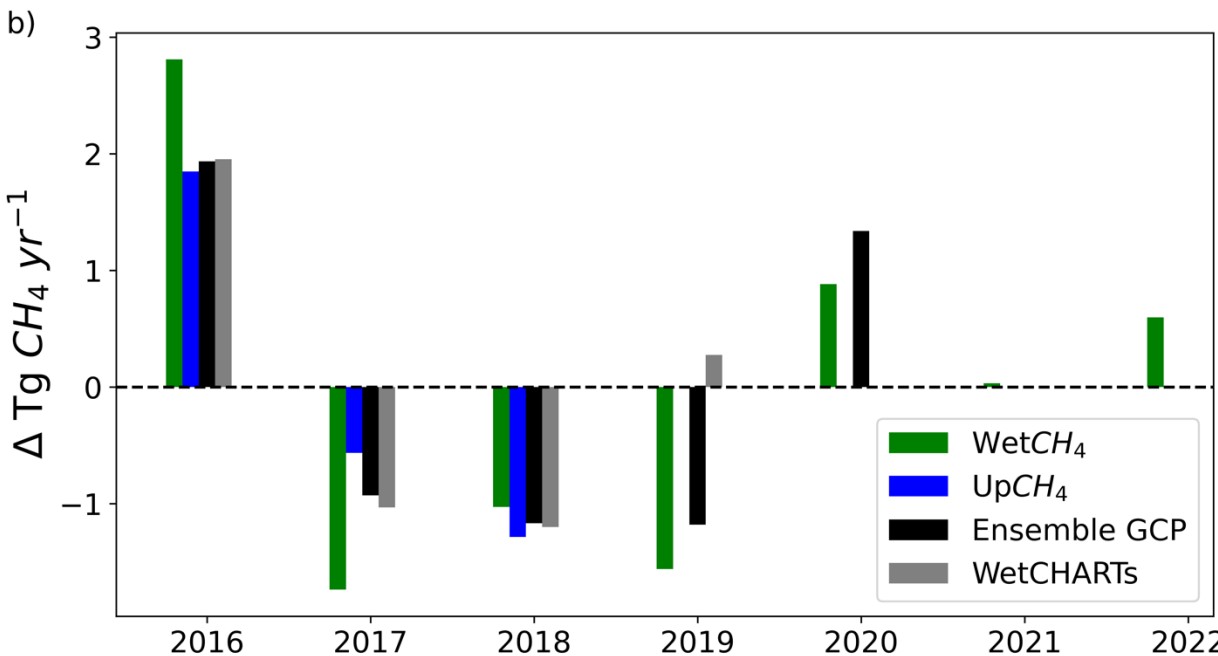

Figure 8 Wetland $CH_4$ a) annual emissions and associated uncertainties in colored shades and
b) variations relative to multi-year means in the research domain (45° - 90° N). Wetland area
data applied in WetCH$_4$ and UpCH$_4$ was WAD2Mv2. Time periods of multi-year means: WetCH$_4$
(2016-2022); UpCH$_4$ (2016-2018); GCP Bottom-up ensemble mean (2016-2020); WetCHARTs
(2016-2019).
Subregional annual emissions and interannual variability (Fig. 9) of WetCH$_4$ were calculated for
eight subregions in the northern high latitudes (Fig. S11): Siberian tundra, East Siberia, West
Siberia, Fennoscandia, Canadian tundra, East Canada, West Canada, and Alaska. The main
differences in WetCH$_4$ estimated emissions between WAD2Mv2 and GLWDv1 occurred in the
East Siberia, East Canada, West Canada, and Alaska subregions. However, interannual
variabilities were similar. Interannual variations from West Siberia accounted for 51% the
variations in domain emissions (Fig. 9a). The positive change in East Canada canceled the
negative change in West Siberia in 2021, resulting in low variability in the domain emission for
that year (Fig. 8). The relative interannual variability, which was calculated as the percentage of
a subregional variation to its period mean, was attributed to those from West Siberia,
Fennoscandia, West Canada, and Alaska (Fig. 9b).

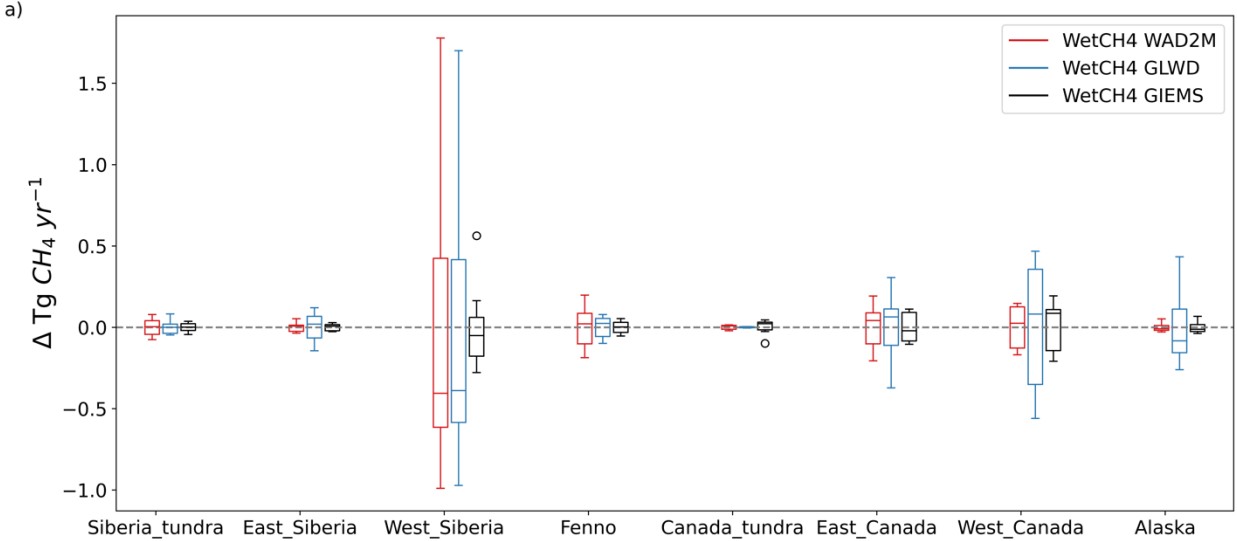

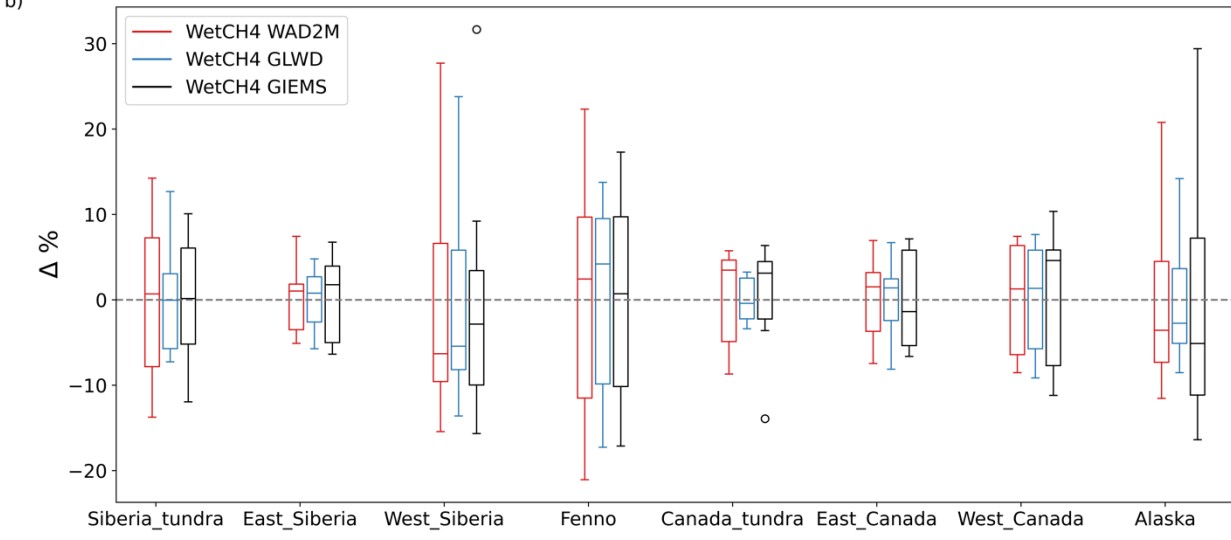

Figure 9 Interannual variations and variability in subregions predicted by WetCH$_4$ with
WAD2Mv2, GLWDv1, and GIEMS2, respectively: (a) interannual variations with respect to
period means (2016-2022); (b) relative variability as the percentage of its period mean. Delta in
the y axis denotes the annual emissions minus mean annual emissions in the period 2016-
2022. The boxplots show the first quartile, the median, and the third quartile of the data with the
whiskers denoting the 1.5x interquartile range below/above the first/third quartile.

# 793    4. Discussion

This study provides new estimates of daily scale 10-km wetland $CH_4$ fluxes for the northern
terrestrial wetland region, upscaled from EC data. The upscaling framework was driven by
MERRA2 meteorological variables and soil temperatures and constrained by satellite products
from SMAP soil moisture and MODIS NBAR, resulting in a good prediction accuracy (mean $R^2$ =
0.70 and mean MAE =27 nmol $m^{-2}$ $s^{-1}$) in monthly mean fluxes. Model agreement worsened at
daily and weekly timesteps due to higher variability in $CH_4$ fluxes at finer temporal resolutions. In
our framework, we applied a rigorous criterion on the counts of half-hourly observations to
control the selection quality of daily gap-filled data, which may filter out errors introduced by the
gap-filling process or lack of observations for calculating daily means. The improvement in
model performance can be partly attributed to the inclusion of soil temperature, satellite
assimilation of soil moisture, and MODIS vegetation reflectance in the framework that
represents controlling factors or proxies of $CH_4$ fluxes recognized in field experiments and
synthesis studies (Fig. 3).

## 808    4.1 Important drivers to improve RF model predictive performance

Soil temperature plays an important role in microbial growth and dormancy (Chadburn et al.,
2020), and exponentially affects microbial $CH_4$ emission rates although the temperature
sensitivity varies across space and time (van Hulzen et al., 1999; Knox et al., 2021). In northern
wetlands, soil temperature is often more spatially variable relative to air temperature due to
snow insulation and active layer depth (Smith et al., 2022; Wang et al., 2016; Yuan et al., 2022),
and thus should be considered in upscaling models. Compared to air temperature or land
surface temperature that were used in previous upscaling studies (McNicol et al., 2023; Peltola
et al., 2019), the inclusion of MERRA2 soil temperatures in WetCH4 likely contributed to a higher
model predictive performance, although the impact of scale mismatch between the native
MERRA2 spatial resolution and the local footprints on the upscaled fluxes were not quantified.
Independent validation studies found significant correlations in the temporal trend and seasonal
cycles between MERRA2 soil temperatures and *in situ* observations (Li et al., 2020; Ma et al.,
2021) in the U.S. and mid-latitude Eurasia. However, lower correlations and overestimated
monthly variability were found in the cold season in Pan-Arctic (Herrington et al., 2022). This
suggests the impact of the uncertainty in MERRA2 soil temperatures were concentrated in the
cold season, when $CH_4$ fluxes were low. The agreement between ensemble means of soil
temperatures from eight reanalysis and land data assimilation system products and station
measurements improved in the pan-Arctic region (Herrington et al., 2022), suggesting the
potential to reduce upscaling uncertainty forced by the ensemble mean of reanalysis datasets.

Emergent vegetation with aerenchyma affects the recent substrate availability and the plant-
mediated transport of $CH_4$ (Kyzivat et al., 2022; Melack and Hess, 2023). We used the full land
bands of the MODIS NBAR product rather than derived vegetation indices used in previous
upscaling studies, as signals indicating wetland vegetation functional characteristics may be lost
when merging bands to derive simple vegetation indices (Chen et al., 2013). In our study, the
near-infrared and shortwave infrared bands (NBAR bands 2, 5, and 7) presented relatively high
importance in the RF model due to their associations with vegetation characteristics and water
table dynamics in northern peatlands (Baskaran et al., 2022; Burdun et al., 2023). Satellite
inputs provide high spatial resolution constraints on the environmental variability and help
reduce model spatial predictive errors (Fig. 3), indicating the requirement of high spatial
resolution driving input for accurately modeling wetland $CH_4$ fluxes (Elder et al., 2021).
Surface and rootzone soil moisture are important controls on ecosystem anaerobic metabolism.
Low soil moisture implies aerobic conditions and allows methanotrophic bacteria to consume
$CH_4$, whereas high soil moisture enables $CH_4$ production and suppresses consumption (Liebner
et al., 2011; Olefeldt et al., 2013; Spahni et al., 2011). Soil wetness estimated in the rootzone
and the profile from SMAP measurements may be able to capture water table dynamics and
hence ranked as important in WetCH4 model performance. Validation of the SMAP level 4 soil
moisture data assimilation product has shown that it meets the performance requirement of
unbiased root-mean-square error <0.04 $m^3/m^3$ (Colliander et al., 2022). However, the validation
sites are mostly located in North American grassland, cropland and shrubland, requiring more *in*
*situ* soil moisture observations in high latitude tundra and peatland. Regional validation studies
suggested uncertainties of satellite derived soil moisture including SMAP at high latitudes were
high (Högström et al., 2018; Wrona et al., 2017) and remained to be addressed.
Underground processes of $CH_4$ production and oxidation are difficult to model (Ueyama et al.,
2023b), especially for seasonal cycles in the northern high latitudes. A hysteresis effect that
manifests intra-seasonal variability in the dependence of $CH_4$ fluxes on temperature has been
observed at EC sites (Chang et al., 2021), but it was not reproduced in WetCH4. Positive
hysteresis and the difference in frozen status from topsoil to deep soil during autumn freeze
results in zero curtain periods that have been observed at high latitude tundra (Bao et al., 2021;
Zona et al., 2016), the occurrence of which was subsequently underestimated in our model.
The amount of additional substrate available for methanogenesis due to soil freezing/thawing,
missing in our framework, could be a controlling factor of the occurrence of this phenomenon.
Higher substrate availability elevates methanogen abundance and activities during autumn
freeze (Bao et al., 2021). However, spatially explicit substrate data are not available. Using
proxies such as net primary production or EVI for substrate availability might be oversimplified
(Larmola et al., 2010; Li et al., 2016; Peltola et al., 2019). In addition, the uncertainty of deep
soil temperature of training inputs in late autumn may hinder the model's ability to capture
patterns of high emissions during zero curtain periods observed at Alaska tundra (Fig. S10).
More temporally accurate soil temperature data is needed to delineate the soil freezing progress
and properly constrain predictions of $CH_4$ emission during the cold season (Arndt et al., 2019).
The UpCH4 results (McNicol et al., 2023) also suggest that simply imposing lags to temporal
predictors in RF cannot capture complex intra-seasonal variability due to the complicated lag
effects interacting with the water table depth (Turner et al., 2021). Without timestamps in
predictors, RF treats time series fluxes independently, which may limit its predictive
performance. Deep learning models designed to account for temporal progress in data, such as
Long Short Term Memory (LSTM) neural networks, may improve modeling accuracy of
seasonal cycles (Reichstein et al., 2019; Yuan et al., 2022).

## 880 4.2 Data limitations in current EC $CH_4$ observations

Data deficiency in EC $CH_4$ flux observations in winter and in under-represented areas limited the
RF model's extrapolation ability. Data abundance and representativeness across space, time,
and wetland types drives model performance and ability to extrapolate for the data-driven
approach. The 26 wetland EC sites included in this study are largely located in Fennoscandia,
East Canada and Alaska (Fig. 2), leaving some regional emission hotspots under-represented.
For instance, Western Siberian Lowlands, the large wetland complex and the major contributor
of interannual variations of $CH_4$ in the region, has little data. The nearest site (RU-VRK, not
included in this study due to the observations before our study period) is situated on the western
side of the Ural Mountains, within the Usa River Depression. Cold season emissions could
contribute a substantial fraction of the Arctic tundra annual $CH_4$ budget (Mastepanov et al.,
2008; Mavrovic et al., 2024; Zona et al., 2016). But after filtering, 23% of the EC data in high
latitudes (>60° N) were recorded between November and March, which could be insufficient for
accurately modeling and upscaling zero curtain period fluxes.

Ten bog and fen sites used for modeling contain all season daily flux records with more than 11
half-hourly observations per day, all from Fennoscandia and Canada. Although Alaska is
represented by 11 wetland sites, sufficient winter observations with good quality are still
needed. West Siberian Lowlands are underrepresented by EC $CH_4$ sites. Missing data in
MODIS NBAR due to snow cover or gaps in SMAP reduced training data by 31% and 48% in
the study domain, respectively. Filling data of MODIS NBAR to account for snow cover
information and gap-filling SMAP soil moisture products can make full use of available EC
observations and help improve model performance in cold seasons. Since gaps in winter SMAP
data were filled with zero values, our approach has limitations in the estimation in winter soil
moisture gaps in areas where zero curtain and talik were not represented by our interpolated
soil temperatures, for example, in coastal areas.

Many wetland sites in the study are located in areas with peatland presence, with 35% of sites
in peatland-rich areas with >50% peatland cover (Hugelius et al., 2020). Mineral soil (soil
containing less than 12% organic carbon by weight) marshes, though covering only 5% of the
total wetland area in the northern high latitudes, need to be considered when deploying new EC
sites due to their high $CH_4$ emissions (Kuhn et al., 2021; Olefeldt et al., 2021). This study
identified regional $CH_4$ emission hotspots and areas undergoing strong interannual variations,
which are yet not part of the current FLUXNET network. However, the 10 km resolution of the
RF estimates prohibits the identification of local hotspots that may occur at <1-10 m scales
(Elder et al., 2021). The wall-to-wall flux maps also provide spatially continuous information for
effectively further developing the $CH_4$ flux tower network.

## 4.3 Budget comparison

WetCH$_4$ estimated annual and seasonal mean emissions that were comparable to existing data-
driven products in the study domain (Table S5). With the dynamic WAD2Mv2 map, our
estimation was 0.7 Tg $CH_4$ yr$^{-1}$ smaller than UpCH$_4$ due to the mean seasonal cycles between
2010 and 2020 from WAD2M applied in our estimation. With the same static GLWDv1 map, our
estimation was about 22% larger than the estimate from Peltola et al. (37.5 ±12 Tg $CH_4$ yr$^{-1}$ for
2013-2014) despite the different periods. This is attributed to higher fluxes estimated by WetCH$_4$
in DJF and JJA seasons. With two versions of the static GLWD maps, we estimated potential
annual emissions between 46.0 and 51.6 Tg $CH_4$ yr$^{-1}$. Compared to GLWDv1, version 2 of
GLWD mapped smaller wetland fractions in the Hudson Bay Lowlands with intense $CH_4$ fluxes
and more wetlands in the northwest of the Ural Mountains, Eastern Siberia, and the Sanjiang
Plain, where $CH_4$ intensities were weaker, resulting in a larger estimate of the annual emission
(Fig. S13). The wide range of data-driven estimates was driven by the differences in wetland
maps. While WAD2M provides crucial information on wetland inundation dynamics controlling
interannual and inter-seasonal changes in $CH_4$ emitting areas, areas with saturated soil in the
Arctic tundra are likely severely underestimated (Fig. 7d), requiring more accurate maps
delineating wet tundra communities at higher spatial resolution (e.g., < 1 km). Incorporating
wetland fractions derived from high-resolution thematic maps (e.g., CALU) can improve the use
of WAD2M in cold regions. Developing/improving higher resolution microwave remote sensing
products capable of tracking dynamic changes in local soil moisture conditions is also needed.
Together, these two components likely currently yield the largest sources of uncertainty in high
latitude terrestrial $CH_4$ budgets.

Bottom-up estimates on wetland $CH_4$ emissions from data-driven, GCP ensemble means and
WetCHARTs are smaller than the top-down CarbonTracker-CH$_4$ estimate on natural microbial
emissions because the latter includes emissions from aquatic systems. Aquatic $CH_4$ emissions
for this region have been estimated at 5.5 Tg $CH_4$ yr$^{-1}$ from rivers and streams (Rocher-Ros et
al., 2023) and 16.6 Tg $CH_4$ yr$^{-1}$ from lakes (Johnson et al., 2022). The total emissions budget for
wetlands and open water, based on this study and the aquatic estimates, are about 44.9 Tg $CH_4$
yr$^{-1}$, which is 4 Tg $CH_4$ yr$^{-1}$ more than the CarbonTracker-CH$_4$ estimate. The amplitudes of
WetCH$_4$ seasonal mean fluxes align with bottom up and top down estimates. Differences in the
seasonal dynamics of wetland maps are the major source of upscaling uncertainty and result in
various uncertainties between regional estimates. While atmospheric inversion models need
bottom-up estimates as priors, data-driven upscaled $CH_4$ products offer alternatives to process-
based estimates to assist with inversion models in regions where data-driven models perform
well (Bloom et al., 2017; Melton et al., 2013).

## 4.4 Future directions

Future development of EC networks in the northern high latitudes is urgently needed to provide additional observations needed to improve model-based upscaling of $CH_4$ flux budgets, and to address current gaps in ecosystem and regional representation. Deploying new sites in under-represented areas will not only benefit flux upscaling efforts but also our understanding of how ecosystem metabolism responds to the changing climate (Baldocchi, 2020; Pallandt et al., 2022; Villarreal and Vargas, 2021). With the availability of long-term predictor variable data, it is possible to expand upscaling frameworks over longer periods (e.g., 2000 to current), when adequate flux observations in 2000-2010 from chambers are compiled, as 96% of the data were recorded after 2010 in FLUXNET-$CH_4$ (McNicol et al., 2023).

Several data products exist for the meteorological predictor variables. Quantifying measurement uncertainties between products of predictor variables and how the uncertainties propagate to upscaling products need to be addressed in future work. The mismatch of spatial scales between tower footprints and predictor variables may cause underestimation of abruptly high fluxes measured at tower landscapes when environmental conditions are averaged over half-degree grids (Chu et al., 2021; McNicol et al., 2023). Therefore, downscaling predictor variables for developing higher-resolution products is needed, especially for the Arctic region where thermokarst development is shaping permafrost landscapes with fragments of wetlands, thermokarst ponds, and forests (Miner et al., 2022; Osterkamp et al., 2000; Wik et al., 2016). For example, Fang et al. (2022) have downscaled global SMAP surface soil moisture to 1-km resolution, and Optical/Thermal and microwave fusion methods have been developed to downscale soil moisture (Peng et al., 2017). Nevertheless, downscaled products for rootzone or profile soil moisture are needed for upscaling $CH_4$ fluxes as are soil temperature products.

Beyond the ML-based upscaling framework, hybrid modeling of the data-driven approach and process-based models is a promising but also challenging direction of future study (Reichstein et al., 2019). One practice constrained regional data-driven fluxes with top-down estimates via auto-learned weights on per pixel fluxes in a region (Upton et al., 2023). Another practice pretrained a time-dependent ML algorithm with initialization from process-based synthetic data and then fine-tuned the model with observations (Liu et al., 2022). Finally, leveraging physical constraints to increase the interpretability of data-driven models and computation efficiency is still an important factor to consider in all hybrid modeling.

# 5. Code and data availability

The daily $CH_4$ flux intensities in the northern wetlands at a spatial resolution of 0.098° x 0.098°and associated uncertainties, along with daily emissions weighted by WAD2M, GIEMS2, and GLWDv1, can be accessed through https://doi.org/10.5281/zenodo.10802153 (Ying et al., 2024). Source code of ML modeling and upscaling is publicly available at https://github.com/qlearwater/WetCH4.git. Half-hourly EC data is available for download at https://fluxnet.org/data/fluxnet-ch4-community-product/ (Delwiche et al., 2021).

# 6. Conclusions

We developed an ML framework (WetCH$_4$) to upscale daily wetland CH$_4$ fluxes of mid-high northern latitudes at 10-km spatial resolution combining EC tower measurements with satellite observations and climate reanalysis. WetCH$_4$ is novel in that it is the first upscaling framework to introduce SMAP soil moisture and MODIS reflectance in modeling wetland CH$_4$ fluxes to improve accuracy (mean $R^2$ = 0.70). The remote-sensing products provided high spatial resolution constraints associated with the abiotic controllers of CH$_4$ fluxes, indicating the importance of using high spatial resolution inputs in models for accurately simulating the spatiotemporally variable CH$_4$ emissions from heterogeneous northern wetland landscapes. The framework highlights the importance of soil temperature, vegetation, and soil moisture for modeling CH$_4$ fluxes in a data-driven approach. Using WetCH$_4$, an average annual CH$_4$ emissions of 22.8 ±2.4 Tg CH$_4$ yr$^{-1}$ with WAD2Mv2 was estimated and ranged between 15.7 ±1.8 Tg CH$_4$ yr$^{-1}$ with GIEMS2 and 51.6 ±2.2 Tg CH$_4$ yr$^{-1}$ with GLWDv2 from vegetated wetlands (>45° N) for 2016-2022, approximately 14-32% of the global wetland CH$_4$ budget (Saunois et al., 2020). Differences in estimates of wetland CH$_4$ emissions due to different wetland maps applied, highlighting the need for high resolution wetland maps and accurate delineation of wet soil dynamics. Emissions were relatively lower in 2017-2019 and intensified in 2016, 2020 and 2022, with the largest interannual variations coming from West Siberia. Spatio-temporal distributions of CH$_4$ fluxes find emission hotspots and regions of intensified interannual variations that are not currently measured with EC. Comparing with current EC sites, we suggest a need for tower observations in wetlands of West Siberia and West Canada and diversified observations across wetland types. More site observations in soil water related variables are needed for improved understanding of flux controls in northern wetland ecosystems. Future wetland CH$_4$ upscaling work could benefit from improved soil moisture products and hybrid modeling.

# Author contributions

QY, ZZ and BP designed the study. QY conducted data processing, model development, and wrote the draft of the manuscript. QY, ZZ, JW, KA, AV, BR, LB, YO, and BP performed data analysis. All authors contributed to the analysis of results and writing of the manuscript.

# Competing interests

The authors declare that they have no conflict of interest.

# Acknowledgements

This work was supported by funding catalyzed by the TED Audacious Project (Permafrost Pathways). Resources supporting this work were provided by the NASA High-End Computing (HEC) Program through the NASA Center for Climate Simulation (NCCS) at Goddard Space Flight Center. We thank Sara Knox and Gavin McNicol for their helpful suggestions in the early stages of the compilation of EC $CH_4$ fluxes and the development of the upscaling product, respectively. Ben Poulter acknowledges support from the NASA Terrestrial Ecology Program. Annett Bartsch was supported by the European Space Agency CCI+ Permafrost and AMPAC-Net projects. Aram Kalhori was supported by the European Union's Horizon Europe program, grant agreement no. 101056848. We thank Frans-Jan Parmentier and the other anonymous reviewer for their helpful reviewing comments to improve the manuscript. We would also like to acknowledge the research teams contributing to the FLUXNET-$CH_4$ and in situ data for this study.

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
