# Peer review of "WetCH4: A Machine Learning-based"

_Earth System Science Data, 2024_

## Author Comment (AC1)

We thank the Referee #2 for the helpful comments and suggestions. They will help improve the manuscript. We will address comments in the order discussed by the Referee. Our responses are in blue.

Wetlands are the largest natural source of global methane ($CH_4$) emissions, but with the largest uncertainty. Ying et al. generated a machine learning based regional (>45) wetland $CH_4$ upscaling dataset. In general, their work is very important, and they provided a new data-driven benchmark dataset constraint by the most eddy covariance observations, with the highest spatial and temporal resolution, compared with previous ML-based wetland $CH_4$ upscaling products. However, there remain many parts that are not clear or rigorous enough. Detailed comments can be seen as follows:

Major comments:

1. For the feature selection part, why did you choose the first 10 variables? Did you test other numbers of input features? In section 3.1.1, you mentioned that using all the variables and using selected 10 variables showed no significant difference in wetland $CH_4$. Is this strategy still reasonable? Maybe the strategy in Peltola et al., 2019 could be helpful. They calculated the feature importance of all the variables, but finally chose four variables, because that group achieved the best performance. (*Peltola, O., Vesala, T., Gao, Y., Räty, O., Alekseychik, P., Aurela, M., ... & Aalto, T. (2019). Monthly gridded data product of northern wetland methane emissions based on upscaling eddy covariance observations. Earth System Science Data, 11(3), 1263-1289.*)

   Responses: Yes, all candidate variables were tested together within the random forest model and we ranked the feature importance of all candidate variables as shown in Fig.S3. The model performance converged with the top 10 most important variables that were selected as input variables, as indicated by the out-of-bag $R^2$ metric and Fig. S3.

   Our modeling framework differed from Peltola et al., 2019 by separating in situ variables and gridded variables and modeling at the site level and grid level respectively. Previous studies trained and validated models with a mix of in situ and gridded variables but performed upscaling with only gridded variables, causing the evaluation metrics not to show the accuracy of upscaling products. In addition, this modeling strategy tended to favor gridded proxies or variables over in situ variables, leading to the real controlling variables of $CH_4$ fluxes not being selected.

   We improved the strategy by first modeling at the site level with only in situ variables that were available at all wetland EC sites. We then used the gridded version of the selected in situ variables. For some missing controlling factors that were not measured across all sites, we further added remote-sensing-based variables or proxies (e.g., SMAP soil wetness, MODIS reflectance) in a forward selection process and demonstrated the improvement in model predictive performance as shown in

Fig. 3a. As a result, the grid-level modeling and evaluation is consistent with the upscaling and reflecting the accuracy of the upscaling product.

We clarified this on lines 192-200 and 369-375 in the clean version of the revised manuscript.

2. The workflow seems a little bit confusing to me. Please feel free to correct me if I misunderstood. It seems that the feature selection only included the variables you get from the MERA2 dataset. Why are the variables from remote sensing dataset excluded in the feature selection step, but directly added into the final RF model? Is that fair to all the variables?

Responses: The feature selection was performed at the site level with in situ measurements. At the grid level, we further performed a forward feature selection by adding remote-sensing datasets to the MERRA2 data. By evaluating the impacts of adding constraints from remote-sensing data on the grid-level model performance as shown in Fig.3a), we prove that adding remote-sensing variables can improve the model's ability to explain the average variability in daily $CH_4$ fluxes across validation sites and reduce prediction errors.

To better explain the framework, we edited sentences on lines 192-200 and 285-395.

3. The final produced dataset is 0.098*0.098degree, but the spatial resolution of input datasets (e.g., MERRA2) is much lower. Similarly, the wetland extent dataset (WAD2M, GIEMS) also has lower spatial and temporal resolution. Will that lead to uncertainties in your final estimation? At least, some discussion of this issue should be added to the manuscript.

Responses: Thanks for this helpful suggestion. We interpolated MERRA2 data to 0.098° x 0.098° weighted by MERIT-DEM. We then modeled and upscaled at this spatial resolution. As a result, the model accuracy metrics reflected a portion of the errors due to the scale difference between MERRA2 input and in situ measurements.

Per your suggestion, we discussed the need for accurate and dynamic wetland maps at high spatial resolution to improve wetland $CH_4$ estimations in the study area. We suggested incorporating wetland fractions derived from high-resolution thematic maps (e.g. CALU) to improve the use of WAD2M in cold regions.

4. L362-364: I think it is not surprising to see that groups (2) and (3) have lower accuracy, because they only contain features from soil wetness or NBAR, but missed the most important information from the features provided by MERRA2 (which you revealed in the feature selection part). Thus, if my understanding is correct, would it be more reasonable to set the input feature as MERRA2, MERRA2 + NBAR, MERRA2+SMAP, and compare them to MERRA2+ all RS data?

Responses: Thanks for the suggestion. We redesigned the model input variable settings to accommodate this suggestion and your comment #6. We updated Fig. 3. Please see our responses to comment #6 for a complete description.

5. Did you test uncertainties from MERRA2? Will the estimation and key findings be the same if using different reanalysis datasets?

Responses: We improved the discussion on the uncertainties in MERRA2 and the potential impacts on emission estimates on lines 791-797:

"However, lower correlations and overestimated monthly variability were found in the cold season in Pan-Arctic (Herrington et al., 2022). This suggests the impact of the uncertainty in MERRA2 soil temperatures were concentrated in the cold season, when $CH_4$ fluxes were low. The agreement between ensemble means of soil temperatures from eight reanalysis and land data assimilation system products and station measurements improved in the pan-Arctic region (Herrington et al., 2022), suggesting the potential to reduce upscaling uncertainty forced by the ensemble mean of reanalysis datasets."

6. Figure 6: I am curious why DEM is the most important feature. You mentioned it highly correlated with air pressure, but the importance of air pressure is very low. Please share more explanation of the mechanisms of how DEM affects wetland $CH_4$.

Responses: Thank you for this helpful comment. We found that the variable importance ranked by the impurity decreases in RF models affected the interpretation of real controlling variables when covariates existed. The collinearity among input variables (such as temperatures at different depths, DEM and air pressure, air temperature) allows some of the removed variable's information to be retained, potentially distorting its true importance. This highlights the need for careful interpretation of correlated features' importance and is the reason why DEM appeared so important in the previous variable importance analysis. To address this,

1, We updated input variables by using interpolated MERRA2 variables (at ~10 km spatial resolution) weighted by DEM for modeling and removing DEM from the input predictors. We clarified this on lines 291-296. Accordingly, we updated Table 1 to reflect this modeling spatial resolution change in MERRA2 data.

2, We improved our design of input feature settings at the grid-level modeling. We first built a baseline grid-level model with independent variables after a pairwise Pearson correlation test (Fig. S14) to exclude covariates. The resulting baseline features included air pressure (pa), latent heat flux (le), sensible heat flux (h), soil temperature (ts2), rootzone soil wetness (sm_r_wetness), slope, spi, and cti. Then we designed four additional different model settings by changing predictor variables, including (1) baseline variables plus covariates, (2) only variables from MODIS NBAR, (3) baseline variables plus NBAR bands, and (4) all predictor variables. In this forward feature selection process, we evaluated the impacts of adding constraint variables from remote sensing products on model performance. RF models can enhance robustness when handling correlated input variables, so these collinear variables shouldn't negatively

affect model performance, only the variable importance assessment. This is due to the RF algorithm randomly selecting subsets of input variables and choosing the best one for splitting nodes during tree constructions. We modified the description on lines 385-295. We demonstrated error reduction and model improvement as new variables, including physically independent MODIS NBAR observations, were added as shown in Fig. 3a.

3, We updated feature importances from the baseline model with non-covariates. We merged Fig. 3 and Fig. 6, and showed the importance of baseline features in Fig. 3b. The new result demonstrated the importance of soil temperature and moisture, as described in the revised manuscript on lines 539-546.

4, We updated the predictive performance metrics of the upscaling model (lines 526-537, 561-592) and ultimately the upscaling results from non-DEM ensemble models (the results section 3.2 Upscaled wetland $CH_4$ emissions). The new model improved performance at wet tundra sites but enlarged errors at a few fen and bog sites. Overall, it slightly overestimated $CH_4$ fluxes at the validation sites as shown by positive bias (mean ME). The upscaling results from the new model manifest slightly higher flux intensities in wet tundra and in the summer season (JJA), resulting in increases in the estimates of mean annual emissions by ~2 Tg $CH_4$ $yr^{-1}$ with WAD2M to ~5 Tg $CH_4$ $yr^{-1}$ with GLWD v1 and v2. No significant change in the absolute and relative interannual variability in subregions.

5, We added a discussion in the supporting materials Text 6 about the impact of elevation on explaining the intra-site variability within the existing wetland sites of northern high latitudes. We tested the impacts of elevation on model performance in explaining the inter-site variability of $CH_4$ upon the current locations of wetland EC sites. We recognized that elevation may act as a factor in discerning fen and bog sites with associated wetland attributes that may not be included by other input variables.

Minor comments:

1. L183: The boundary of Arctic-boreal is not exactly the same as '>45 degree'. If your final dataset is >45 area, you cannot say it is Arctic-boreal region. Similar problems appeared several times in the manuscript. Please go through the whole paper and correct them.

   Responses: Thanks for this suggestion. We have rephrased the study area to wetland >45° N.

2. Are the important features the same at different sites? Or are they the same across different wetland types? Did you build separate models for different wetland types? Or use one model for all types?

Responses: We built one model for all types of wetlands as the gridded wetland fraction information by wetland types was not available in the whole study area that was required for wetland fluxes upscaling.

3. Vegetation activity showed significant impacts on wetland CH4 emissions in many previous studies, especially in the northern wetlands. Why not include proxies of vegetation (such as, LAI, GPP, …) into your feature selection?

Responses: Previous studies (Peltola et al., 2019; McNicol et al., 2023) evaluated MODIS-derived EVI as a proxy for GPP as a candidate predictor. However, none of them selected EVI (concurrent or lagged) in their upscaling models because the inclusion of EVI did not improve the model performance as much as the temperature-related variables did. Therefore, we did not directly include proxies of vegetation productivity in the feature selection, instead, we included constraints from MODIS reflectance bands that were used to produce GPP, EVI, or LAI as well as surface water indices. We explained this on lines 335-337.

4. Figure 4: Why is monthly prediction much better than that of daily prediction, especially in terms of $R_2$? Please add more explanation to the manuscript.

Responses: Model predictive performance on aggregated monthly means of $CH_4$ fluxes increased by 37% as compared to daily means ($R^2 = 0.70$, Fig.4, Table S4). Model agreement worsened at daily and weekly timesteps due to higher variability in $CH_4$ fluxes at finer temporal resolutions (lines 768-769). The amount of noise in the flux data is much higher at a daily resolution while mean monthly fluxes smooth some of this noise away.

5. Figure 7: For carbon-tracker, why did you use natural microbial emissions instead of wetland emissions? It seems that carbon-tracker also has an output layer of wetland CH4.

Responses: Natural microbial emissions primarily comprise wetland and aquatic emissions in northern high latitudes. According to Oh et al., 2023, the aquatic $CH_4$ sources were not discerned from wetland emissions in the current release of CarbonTracker-$CH_4$. Therefore, we used natural microbial emissions.

Reference: Youmi Oh, Lori Bruhwiler, Xin Lan, Sourish Basu, Kenneth Schuldt, Kirk Thoning, Sylvia E. Michel, Reid Clark, John B. Miller, Arlyn Andrews, Owen Sherwood, Giuseppe Etiope, Monica Crippa, Licheng Liu, Qianlai Zhuang, James Randerson, Guido van der Werf, Tuula Aalto, Stefano Amendola, Sébastien C. Andra, Marcos Andrade, Nhat A. Nguyen, Shuji Aoki, Francesco Apadula, Ikhsan B. Arifin, Sabrina Arnold, Mikhail Arshinov, Bianca Baier, Peter Bergamaschi, Tobias Biermann, Sebastien C. Biraud, Pierre-Eric Blanc, Gordon Brailsford, Huilin Chen, Aurelie Colomb, Cedric Couret, Paolo Cristofanelli, Emilio Cuevas, Lukasz Chmura, Marc Delmotte, Lukas Emmenegger, Gulzhan Esenzhanova, Ryo Fujita, Luciana Gatti, Elise-Andree Guerette, László Haszpra, Michal Heliasz, Ove Hermansen, Jutta Holst, Tatiana Di Iorio, Armin Jordan, Müller-Williams Jennifer, Anna Karion, Teruo

Kawasaki, Victor Kazan, Petri Keronen, Seung-Yeon Kim, Tobias Kneuer, Katerina Kominkova, Elena Kozlova, Paul Krummel, Dagmar Kubistin, Casper Labuschagne, Ray Langenfelds, Olivier Laurent, Tuomas Laurila, Haeyoung Lee, Irene Lehner, Markus Leuenberger, Matthias Lindauer, Morgan Lopez, Reza Mahdi, Ivan Mammarella, Giovanni Manca, Michal V. Marek, Martine D. Mazière, Kathryn McKain, Frank Meinhardt, Charles E. Miller, Meelis Mölder, John Moncrieff, Heiko Moosen, Caisa Moreno, Shinji Morimoto, Catherine L. Myhre, Alberth C. Nahas, Jaroslaw Necki, Sylvia Nichol, Simon ODoherty, Nina Paramonova, Salvatore Piacentino, Jean M. Pichon, Christian Plass-Dülmer, Michel Ramonet, Ludwig Ries, Alcide G. di Sarra, Motoki Sasakawa, Daniel Say, Hinrich Schaefer, Bert Scheeren, Martina Schmidt, Marcus Schumacher, Mahesh K. Sha, Paul Shepson, Dan Smale, Paul D. Smith, Martin Steinbacher, Colm Sweeney, Shinya Takatsuji, Gaston Torres, Kjetil Tørseth, Pamela Trisolino, Jocelyn Turnbull, Karin Uhse, Taku Umezawa, Alex Vermeulen, Isaac Vimont, Gabriela Vitkova, Hsiang-Jui (Ray) Wang, Doug Worthy, Irène Xueref-Remy. CarbonTracker CH4 2023, 2023.DOI: 10.25925/40jt-qd67

6. What GCP models did you include in comparison? All the top-down and bottom-up models in Saunois et al., 2019? It would be better to give more information of what model did you used in the supplementary. Or at least, add the citation of GCP models.

   Responses: Thanks for the helpful suggestion. We added information about the GCP models in the supporting materials:

   The bottom-up estimates we used for comparison were from sixteen wetland $CH_4$ models ($CH4MOD_{wetland}$, CLASSIC, DLEM, ELM-ECA, ISAM, JSBACH, JULES, LPJ-MPI, LPJ-wsl, LPJ-GUESS, LPX-Bern, ORCHIDEE, SDGVM, TEM-MDM, VISIT, TRIPLEX-GHG) in the Global Carbon Project (GCP) Methane Budget (Z. Zhang et al., 2024).

7. Figure 10: Why exclude WetCH4-GIEMS?

   Responses: We updated the figure by adding interannual variability estimated in WetCH4-GIEMS.

8. Figure 8d: Please give more description of land and CALU data, and explain how you generate wetCH4-land and wetCH4-CALU, and why did you use them.

   Responses: We modified the manuscript as below:

   "Given that the wetland area in this region is uncertain (Miller et al., 2016), we computed mean seasonal cycles over the land assuming all land in this area is water saturated in the soil, over freshwater wetlands of CALU, and over WAD2M and Hydrolakes, representing three different scenarios. In the lowland area of the North Slope (74295 $km^2$ spanning between 69.8°N - 71.4°N, 164.4°W - 152.7°W), the wetland area was estimated at 10611 $km^2$ from CALU, 4800 $km^2$ from GLWDv2, and 4049 $km^2$ from the maximum extent month in July of WAD2Mv2, respectively."

9. Add citations: L78-80, L99-104.

   Responses: Thanks for your suggestion. We added citations:

   "The uncertainties in the estimates of wetland $CH_4$ emissions are primarily attributed to challenges in mapping vegetated wetlands versus open water leading to double counting (Thornton et al., 2016), seasonal wetland dynamics and uncertainties in estimates on flux rates."

   "Field observations of gas fluxes typically measure $CH_4$ exchange between the land and atmosphere at sub-meter to ecosystem (100s of m to km) scales (Bansal et al., 2023; Chu et al., 2021)."

10. L921-928: Font style.

    Responses:

    Thanks for pointing this out. We edited the Font in this paragraph.

---

## Author Comment (AC2)

We thank the Referee #1 for the helpful comments and suggestions. They will help improve the manuscript. We will address comments in the order discussed by the Referee. Our responses are in blue.

The study by Ying et al. sets out to upscale methane fluxes across the northern high latitudes (>45° N) with the use of machine learning (i.e., random forest). While there have been recent studies with similar approaches, this study is a useful addition to those existing ones, exploring some new directions. There is a lot of detail here, and I appreciate that the authors try to evaluate their results using different wetland maps.

That being said, the paper still needs quite some improvement. The writing is sometimes hard to follow, or imprecise, and this should be improved. I have suggested a large number of fixes down below, but it would be good if the language of the whole paper is checked thoroughly.

I was also surprised to see DEM coming on top as the most important variable in the LOOCV scheme (Fig. 6), which doesn't make sense to me. If I understand correctly, then DEM refers here only to elevation (since slope, spi and cti are defined separately). It is not explained properly how elevation would influence methane emissions. Temperatures become lower and precipitation increases with altitude, but temperature and wetness are already included as variables and score lower in this scheme. Is DEM simply a good predictor because most wetlands are found at low elevations rather than that it's a driver of emissions? I would like to see a better explanation for this result, and evidence that it's not an artificial signal.

Responses:

Thank you for this helpful comment. We agree that elevation should not be considered an ecological controlling factor for wetland $CH_4$ fluxes as it covaries with other meteorological variables. We found that the variable importance ranked by the impurity decreases in RF models affected the interpretation of real controlling variables when covariates existed. The collinearity among input variables (such as temperatures at different depths, DEM and air pressure, air temperature) allows some of the removed variable's information to be retained, potentially distorting its true importance. This highlights the need for careful interpretation of correlated features' importance and is the reason why DEM appeared so important in the previous variable importance analysis. To address this,

1, We updated input variables by using interpolated MERRA2 variables (at ~10 km spatial resolution) weighted by DEM for modeling and removing DEM from the input predictors. We clarified this on lines 291-296. Accordingly, we updated Table 1 to reflect this modeling spatial resolution change in MERRA2 data.

2, We improved our design of input feature settings at the grid-level modeling. We first built a baseline grid-level model with independent variables after a pairwise Pearson correlation test (Fig. S14) to exclude covariates. The resulting baseline features included air pressure (pa), latent heat flux (le), sensible heat flux (h), soil temperature (ts2), rootzone soil wetness

(sm_r_wetness), slope, spi, and cti. Then we designed four additional different model settings by changing predictor variables, including (1) baseline variables plus covariates, (2) only variables from MODIS NBAR, (3) baseline variables plus NBAR bands, and (4) all predictor variables. In this forward feature selection process, we evaluated the impacts of adding constraint variables from remote sensing products on model performance. RF models can enhance robustness when handling correlated input variables, so these collinear variables shouldn't negatively affect model performance, only the variable importance assessment. This is due to the RF algorithm randomly selecting subsets of input variables and choosing the best one for splitting nodes during tree constructions. We modified the description on lines 385-295. We demonstrated error reduction and model improvement as new variables, including physically independent MODIS NBAR observations, were added as shown in Fig. 3a.

3, We updated feature importances from the baseline model with non-covariates. We merged Fig. 3 and Fig. 6, and showed the importance of baseline features in Fig. 3b. The new result demonstrated the importance of soil temperature and moisture, as described in the revised manuscript on lines 539-546.

4, We updated the predictive performance metrics of the upscaling model (lines 526-537, 561-592) and ultimately the upscaling results from non-DEM ensemble models (the results section 3.2 Upscaled wetland $CH_4$ emissions). The new model improved performance at wet tundra sites but enlarged errors at a few fen and bog sites. Overall, it slightly overestimated $CH_4$ fluxes at the validation sites as shown by positive bias (mean ME). The upscaling results from the new model manifest slightly higher flux intensities in wet tundra and in the summer season (JJA), resulting in increases in the estimates of mean annual emissions by ~2 Tg $CH_4$ yr$^{-1}$ with WAD2M to ~5 Tg $CH_4$ yr$^{-1}$ with GLWD v1 and v2. No significant change in the absolute and relative interannual variability in subregions.

5, We added a discussion in the supporting materials Text 6 about the impact of elevation on explaining the intra-site variability within the existing wetland sites of northern high latitudes. We tested the impacts of elevation on model performance in explaining the inter-site variability of $CH_4$ upon the current locations of wetland EC sites. We recognized that elevation may act as a factor in discerning fen and bog sites with associated wetland attributes that may not be included by other input variables.

Other than that, I have some comments about definitions. First of all, the paper mentions a few times that it aims to be a study of the Arctic-Boreal region, while in fact it looks at the whole region north of 45° N and includes two sites from northern Germany, which are clearly outside of the Arctic-Boreal region. I see from Table S2 that multiple sites in Canada and the USA are also classified as temperate. Either restrict your domain to the Arctic-Boreal region or rephrase in the document that you are looking at northern high latitudes. In that case, please add information on methane emissions in temperate biomes to the introduction.

Responses: Thanks for your suggestion. We have rephrased the "Arctic-Boreal region" to "northern high latitudes" in the manuscript. We added mean methane fluxes in temperate biomes in the introduction.

Finally, I understand the use of WAD2M, and this manuscript covers some of its limitations, but this is a missed opportunity to improve the applicability of this product for high latitude wetlands. Correct me if I'm wrong but WAD2M shows a seasonal cycle, going towards zero where soils are frozen and underestimating ecosystems such as bogs where methane emissions occur also when the soil is not inundated. Also, northern wetlands are rather stable, in contrast to wetlands at lower latitudes, and observations show that methane can still be emitted in winter. So, a seasonal cycle in wetland extent is not that useful for these northern environments.

The authors have taken the average seasonal cycle for WAD2M, but this does not solve this problem. In fact, it may introduce new problems since you use SMAP soil wetness, which will correlate with the inundation dynamics in WAD2M. So why not keep wetland extent from WAD2M fixed throughout each year, for example by taking the maximum annual extent, and then model methane emissions according to your observations of soil moisture and other variables? Other solutions may be possible, and the authors acknowledge that WAD2M is not perfect (give the comparison to CALU on the North Slope). Since WAD2M is being used by many people in the community, I would have liked to have seen a discussion on how to improve its usefulness for cold climates from this paper.

Responses:

Thanks for pointing this out. Per your suggestion, we calculated the mean seasonal cycle of $CH_4$ fluxes within the maximum annual extent of WAD2M to separate the compound impacts of seasonal changes in WAD2M wetland extent and in flux intensities that were already affected by SMAP soil wetness.

We added seasonal cycle plots weighted by the annual maximum extent of WAD2M for 2016-2020 in subplot c). This can help separate the impacts of seasonal WAD2M on the seasonal cycles of $CH_4$ emissions from those of modeled flux rates. Please refer to:

Fig. 7 Multi-year average seasonal cycles of wetland $CH_4$ emissions: (a) comparison of ML upscaled mean seasonal cycles in reference wetland areas (WAD2Mv2) with the cycles from process-based models in the northern mid-high latitudes (45° - 90° N); (b) same comparison for northern high latitudes (60° - 90° N) and addition of atmospheric CarbonTracker-$CH_4$ attributed microbial emissions (2016-2022); (c) comparison of three ML upscaled mean seasonal cycles of $CH_4$ emissions with different wetland area maps (WAD2Mv2, WAD2Mv2 maximum extent, GIEMS2, GLWDv1); (d) comparison of WetCH4 mean seasonal cycles over the land (black line), weighted by wetland of the CALU map (olive line), or weighted by fractions of WAD2Mv2 (green line), with estimates of $CH_4$ fluxes in growing seasons from CARVE retrievals in North Slope area of Alaska (Zona et al., 2016).

We added the following sentences to the results: "We decoupled the mean annual seasonal cycle for WAD2M from the emission seasonality by using a fixed maximum WAD2M extent. The resulting seasonal emissions primarily driven by soil temperatures and moisture manifested elevated emissions in all months and an intensified seasonal cycle."

We also added in the discussion: "Incorporating wetland fractions derived from high-resolution thematic maps (e.g., CALU) can improve the use of WAD2M in cold regions."

Detailed comments:

Line 79-80: please reference Thornton et al. (2016) who originally raised this issue of double counting:

Thornton, B. F., Wik, M., & Crill, P. M. (2016). Double counting challenges the accuracy of high latitude methane inventories. Geophysical Research Letters, 43(24), 12,569-12,577. https://doi.org/10.1002/2016GL071772

Responses:

Thanks for your suggestion. We added the reference for double counting.

Line 81-89: This paragraph feels more like a list of bullet points rather than text. Please rewrite this to make it more readable.

Responses:

We rewrote this paragraph to provide an introduction to wetland types in northern high latitudes with a focus on the ecosystems and flux rates:

"Characterized by nutrient, moisture and hydrodynamic conditions, northern freshwater wetlands are classified as wet tundra in treeless permafrost areas, peat-forming bogs and fens in boreal biomes, and permafrost bogs (Olefeldt et al., 2021; Kuhn et al., 2021). Bogs were estimated to cover the largest area (1.38-2.41 million km$^2$) in the boreal-Arctic region, followed by fens (0.76-1.14 million km$^2$) and wet tundra (0.31-0.53 million km$^2$) (Olefeldt et al. 2021). Climate change poses significant threats to these wetlands, affecting their extent and the duration of conditions suitable for wetland formation in permafrost zones (Avis et al., 2011). Distinct $CH_4$ fluxes have been observed from wet tundra (Fig. S4, mean ± standard deviation: 13 ±14 nmol m$^{-2}$ s$^{-1}$), bogs (22 ±26 nmol m$^{-2}$ s$^{-1}$) and fens (56 ±88 nmol m$^{-2}$ s$^{-1}$). The rates of $CH_4$ emissions may increase at a faster pace because of intensified warming in the Arctic (Masson-Delmotte et al., 2021; Rawlins et al., 2010; Rößger et al., 2022; Walsh, 2014; Z. Zhang, Poulter, et al., 2023)."

Line 82: "to wet tundra" should be "as wet tundra"

Responses:

Revised as suggested. Thanks!

Line 83: which exceptions?

Responses:

We revised our manuscript in line xx:

"Characterized by nutrient, moisture and hydrodynamic conditions, northern freshwater wetlands are classified as wet tundra in treeless permafrost areas, peat-forming bogs and fens in boreal biomes, and permafrost bogs (Olefeldt et al., 2021; Kuhn et al., 2021)."

Line 91-92: The wording "recent increase" suggests systematic change (i.e. a trend), but you talk only about the difference between 2019 and 2020. That's interannual variability. Please rephrase.

Responses:

We revised the sentence as follows:

"Northern wetlands may account for a portion of the exceptional global surface emissions in 2020 relative to 2019 (6.0 ± 2.3 Tg $CH_4$ $yr^{-1}$) (S. Peng et al., 2022; Z. Zhang, Poulter, et al., 2023)."

Line 104: "half hourly" should be "at half-hourly intervals".

Responses:

Thanks for pointing out. We revised the manuscript per your suggestion.

Line 107: "outside the network": please change to "outside of the network".

Responses:

Thank you and we revised the manuscript per your suggestion.

Line 113: Independent to what?

Responses:

We revised this sentence as follows:

"Data-driven upscaling with empirical models (Bodesheim et al., 2018; Jung et al., 2011), including machine learning (ML) approaches, to compute $CH_4$ fluxes provide independent estimates to those

from process-based models and atmospheric inversions (Bergamaschi et al., 2013; Spahni et al., 2011)."

Line 132: how would this approach lead to a bias? And in which direction?

Responses:

This approach uses highly synthesized flux intensities and lacks spatial and temporal variability in methane fluxes, which could lead to overestimates or underestimates. More importantly, this approach could not validate the generalized flux intensity per land-cover class and estimate a bias.

Line 141: "for the computation efficiency": change to "for computation efficiency"

Responses:

Thank you! We corrected this sentence in the manuscript per your suggestion.

Line 163: The word "freshwater" can be removed here.

Responses:

Thank you! We removed "freshwater" in this sentence per your suggestion.

Line 174: "ensembled" should be "ensemble"

Responses:

Thank you! We revised the manuscript per your suggestion.

Line 187-188: please check the structure of this sentence, right now it's rather confusing.

Responses:

We restructured this sentence as follows:

"The scalable framework of upscaling $CH_4$ fluxes from EC observations (referred to as $WetCH_4$ hereafter), which selects predictors at site level and constructs upscaling models at grid level, is illustrated in Fig. 1."

Line 193: "model RF": do you mean "model with RF"?

Responses:

Yes, we corrected this in the manuscript. Thanks!

Line 201: Don't you mean latitudes? Longitudinal width of grid cells becomes smaller the further North you get, but they stay the same width along latitudes.

Responses:

Yes, we corrected it. The latitudinal segments of grid cells are the same. The longitudinal width gets smaller towards the North Pole.

Fig 2: It's very difficult to see from this map how many EC-towers you used. I count 19 circles, but the text mentions 26 sites? Also, please change the color for wet tundra. Very hard to distinguish this from the wetland fraction.

Responses:

Thanks for your helpful comments. We improved Figure 2 by displacing clustered sites apart and changing the symbol color for wet tundra:

[Figure]

EC tower site-years | EC tower classes | WAD2M

EC tower site-years
- · 1 - 3
- ○ 3 - 5
- ○ 5 - 7
- ○ 7 - 9
- ○ 9 - 11

EC tower classes
- ○ Wet tundra
- ○ Bog
- ○ Fen

WAD2M
wetland fraction
0.930723
0

0    1,000    2,000 km

Line 219: which 8 sites? Please mention the Fluxnet codes here.

Responses:

The 8 sites include US-ATQ, US-BEO, US-BES, US-BRW, US-IVO, US-NGB, US-NGC, US-UAF. We added this information in the revised manuscript.

Line 220: which 4 sites? Again, please mention with Fluxnet codes.

Responses:

The 4 sites are CA-ARB, CA-ARF, CA-PB1, CA-PB2. We added this information in the revised manuscript.

Line 22: "largest high latitude EC-data compilation": please add "for methane. There are larger syntheses for $CO_2$.

Responses:

Thanks for this suggestion. We added it to the sentence.

Line 238-239: Only 2.5% of sites had winter data? So only one in 40? Of your 26 sites?

Responses:

The daily data in winter (DJF) contains 317 entries after quality filtering, accounting for 2.5% of the total 12,784 daily data entries. We revised this sentence to reduce confusion that may cause to the audience:

"As a result, we identified 12,784 daily data entries for upscaling models (Table S2), spanning 2015-2021 with seasonal observation distributions of 44.0% in June-July-August (JJA), 29.0% in March-April-May (MAM), 24.5% in September-October-November (SON), and 2.5% in December-January-Feburary (DJF) (Fig. S2)."

Line 248-249: This is a missed opportunity! Water table depth is a much better predictor than soil moisture. Why not run the site-level model for the subset of sites with water level data?

Responses:

Thanks for pointing this out. Unfortunately, we don't have enough water table depth (WTD) data in the current collection to perform a holistic analysis of the variable importance of WTD on CH4 fluxes over all wetland sites and wetland types.

Upon further analysis, we found that water table depth (WTD) in site-level modeling did not enhance model performance in bogs and fens (comparable out-of-bag $R^2$ and RMSE). Adding soil water content (SWC) slightly improved the model's ability to explain spatiotemporal variability (larger out-of-bag $R^2$) and reduced prediction errors (smaller RMSE).

To compare WTD, we examined WTD and SWC observations in our dataset. The 9 wetland sites that had WTD data included (CASCB, DEHTE, DEZRK, FISI2, FISII, SEDEG, SEST0, USLOS, USUAF), only cover bogs and fens. There are also 9 sites with SWC data (CASCC, FISII, SEDEG, USATQ, USBES, USBZB, USBZF, USICS, USUAF) covering all three wetland types.

We tested site-level modeling with our existing candidate variables and additional WTD or SWC at the sites where WTD or SWC observations were available, respectively. The table below summarizes the model performances:

| site-level | oob r2 | RMSE (nmol CH4 m-2 s-1) |
| --- | --- | --- |
| Existing input for all wetland sites | 0.73 | 30.43 |
| Existing input for all bog sites | 0.85 | 7.2 |
| Existing input for all fen sites | 0.84 | 27.7 |
| Existing input for sites with WTD | 0.84 | 29.29 |
| Adding WTD for sites with WTD | 0.84 | 29.04 |
| Existing input for sites with SWC | 0.83 | 8.37 |
| Adding SWC for sites with SWC | 0.85 | 7.96 |

With this analysis, we modified our manuscript as below:

"We were unable to include water-table depth (WTD) or soil water content (SWC) in our site-level model as they were not available at many sites. However, we tested site-level modeling on candidate predictors with WTD or SWC at the sites where these variables were available (see Supporting Materials Text 2 for more details)."

We added a paragraph in the Supporting Materials Text 2:

"We examined water table depth (WTD) and soil water content (SWC) observations in our dataset. There are 9 wetland sites with WTD data (CA-SCB, DE-HTE, DE-ZRK, FI-SI2, FI-SII, SE-DEG, SE-ST0, US-LOS, US-UAF) covering bogs and fens. There are also 9 sites with SWC data (CA-SCC, FI-SII, SE-DEG, US-ATQ, US-BES, US-BZB, US-BZF, US-ICS, US-UAF) covering all three wetland types. We tested site-level modeling on our existing candidate variables and additional WTD or SWC at the sites where WTD or SWC observations were available, respectively. Adding WTD did not enhance model performance in bogs and fens. Adding SWC slightly improved the model's ability to explain spatiotemporal variability and reduced prediction errors."

Line 263: "modeling": Grid-level or site-level?

Responses:

We added 'grid-level' in the sentence for clarification:

"Daily time series of the nearest 0.5° grid to each EC location were extracted for grid-level modeling."

Line 264: how was this interpolation done? Does it use spatial data with a higher resolution?

Responses:

We interpolated MERRA2 data using the bilinear interpolation method weighted by 0.098° DEM that was averaged from 30m MERIT-DEM data. We modified the sentence:

"The MERRA2 data was further bilinearly interpolated from 0.5° to 0.098° grids weighted by higher resolution elevation data (see information on the digital elevation model below) for the 10-km upscaling products."

Line 273: Did you check with MERRA that temperatures were below freezing for these gaps? That you're relatively confident that soils were indeed frozen? Might not be true for near-coastal areas.

Responses:

Thanks for mentioning this. The interpolated MERRA soil temperatures may not represent local areas of zero curtain and talik, or coastal areas, where gaps in winter SMAP could be filled with some values rather than zero. We added this limitation on data preprocessing in the revised manuscript:

"Gaps in winter SMAP data were filled with zero values to represent frozen soils for upscaling. This may limit the estimation in winter soil moisture gaps where local areas of zero curtain and talik were likely not represented by our interpolated soil temperatures, for example, in coastal areas."

Line 284: This is a very deep root zone for the Arctic! Especially in wetlands that are dominated by sedges, rushes and grasses or in areas where the active layer never becomes deeper than half a meter or less. A root zone of 15 or 20 cm makes much more sense for high latitude wetlands. Maybe SMAP only gives this for the top meter? How does this affect your results?

Responses:

We used the SMAP level-4 assimilation product which was an outcome of assimilating SMAP L-band brightness temperature observations into the Catchment land surface model in an ensemble-based algorithm. Soil moisture and related land surface variables were reportedly estimated at the surface (0-5 cm) and root zone (0-100 cm) globally and validated at 43 core validation sites and 406 network sites (Reichle et al., 2017). The status of frozen ground was screened. Therefore, the assimilated values may reflect the status of a shallow active layer in the Arctic wetlands. Given that root zone soil wetness is an important predicting variable in our model, the variations in active layer depth during the freezing/thawing period over permafrost

regions can result in higher uncertainty in the root zone soil moisture estimates and therefore in the downstream upscaled fluxes.

Line 293: "existing upscaling models": which?

Responses:

The existing upscaling models are listed in Table S1. We revised the manuscript to refer the audience to the table.

Line 305: "a multiple" should be "the multiple".

Responses:

Thanks for pointing this out. We revised the manuscript per your suggestion.

Line 308: "based" should be "based on"

Responses:

Thanks for catching this. We corrected the phrase in our manuscript per your suggestion in line xx.

Line 331: "Rice paddies"? Or "a rice paddy"?

Responses:

We change it to "rice paddies" in the manuscript.

Line 411: Resampled how?

Responses: All wetland maps were resampled to 0.098° x 0.098° resolution with a conservative remapping method for producing the emission products.

Line 429-430: What is the motivation to say that GWLD is the maximum potential emission surface? There are other wetland maps out there. Does GWLD have the highest extent of them all?

Responses: According to Zhang et al. (2021), GWLD has the highest extent of five static global wetland maps. We modified this sentence to:

"Level 3 1-km resolution map excluding classes of lakes, rivers, and reservoirs (Lehner & Döll, 2004) was included to quantify the upper limit of wetland cover."

Line 446: "thought to be": by whom? And if GWLD underestimates in the north slope, then it is clearly not the maximum potential emission surface. Perhaps it underestimates elsewhere also? Can you show a comparison of CALU to WAD2M and GWLD for the North Slope?

Responses: We calculated the wetland area in the lowland of the North Slope (spanning between 69.8°N - 71.4°N, 164.4°W - 152.7°W) from CALU (10611 km$^2$), the maximum extent month in July of WAD2Mv2 (4049 km$^2$), and GLWDv2 (4800 km$^2$). The Circum-Arctic Landcover Units produced unprecedented detail in wetland and vegetation distribution at 10m spatial resolution. Global wetland maps such as WAD2M and GLWD were created at coarser resolutions and represented wetland dynamics/status over a longer period. We agree that GLWDv2 may not accurately capture all potential emitting surfaces.

We modified this sentence in our manuscript as below:

"This regional wetland map was applied for CH$_4$ emission estimation in the North Slope region in Alaska to assess the impacts of different wetland maps on emission estimates in this area when compared against airborne measurements."

We also added the sentence below in the results section:

"In the lowland area of the North Slope (74295 km$^2$ spanning between 69.8°N - 71.4°N, 164.4°W - 152.7°W), the wetland area was estimated at 10611 km$^2$ from CALU, 4800 km$^2$ from GLWDv2, and 4049 km$^2$ from the maximum extent month in July of WAD2Mv2, respectively."

Line 521-525: if these mismatches are due to scale discrepancies, then why would this differ among ecotypes? In particular, why is it so large for fens?

Responses: We modeled the random forest regression across wetland types (or ecotypes), the limitation of which, as we mentioned in the manuscript, tended to underestimate the high flux type (e.g. fens) and overestimate the low flux types (e.g. wet tundra and permafrost bogs). The scale discrepancies would exaggerate this tendency and lead to further underestimation for fens.

Line 543-544: So, this site cannot be well represented by WAD2M since it assumes inundation?

Responses: The mean inundation fraction of the grid where the US-UAF site sits is 2%, which averages the status of wetland cover despite local variations. Therefore, we used all sites locally identified as wetlands at the stage of model development regardless of wetland fractions from coarse-resolution wetland maps.

Figure 4: Please add that these are grid-level model simulations.

Responses:

We edited the caption of Figure 4 as follows:

"Model predictive performance evaluation on RF modeled $CH_4$ fluxes at grid level and independent validations: (a) boxplots of $R^2$, MAE, and RMSE across validation sites by wetland types with mean values denoted in green triangles; (b) pooled daily means density scatter plot; (c) pooled monthly means density scatter plot."

Table 2: What is the value of showing this in the main document? Can it be moved to the supplemental?

Responses: We moved the table to the supplementary.

Line 580 and Figure 6: I don't understand why DEM comes out as a good predictor. Elevation does not control methane emissions. It influences precipitation and temperature, but you already have those variables included.

Responses: We agree that elevation would not control methane emissions. We added experiments and modified our manuscript as described in our responses to the main comment above.

Line 605-606: It's no surprise that these spatial patterns are similar, since the models from the GCP use the same wetland map as you do!

Responses:

We agree with this point since we used the diagnostic model runs of GCP-$CH_4$ that incorporated prescribed wetland extents from WAD2M into the models.

Line 612: Why use monthly inundation data? Wetland extent at high latitudes does not vary much over the year (unlike tropical wetlands). Aren't you enforcing a seasonal cycle that could be better simulated by using temperature?

Responses: We agree that compared to tropical wetlands, dynamics in wetlands extent of high latitudes may show less interannual variability, and we used period means of monthly inundation of WAD2Mv2 and GIEMS2. Temperature is a driving factor of the seasonal cycle in flux intensities. Seasonal wetland extent is another factor that affects the seasonal dynamics in domain emissions in addition to the flux intensity. We modified this sentence to clarify we used mean monthly inundation data:

"Mean monthly wetland inundations are provided by WAD2Mv2 and GIEMS2, which set the dynamic limits for the wetland boundaries of the $CH_4$-emitting surface."

Line 667-668: And large bursts of methane, see the paper by Mastepanov et al. that you cite later on.

Responses: We modified this sentence in our manuscript to:

"Reported emissions (Zona et al., 2016) and large bursts (Mastepanov et al., 2008) from the freezing active layer at permafrost areas in October (zero-curtain period) may not be well captured by our ML model."

Line 689: Why would emissions double due to permafrost thaw? Very speculative, also because permafrost thaw changes the hydrology of the landscape. If this leads to more drainage, then it can lower emissions!

Responses: We agree that future emissions will depend on the state of local hydrology. If permafrost thaw leads to more drainage and less wetland area, emissions could be reduced. We revised the sentences as below:

"Remarkable increases could be in summer and winter if we assume wetland over this region, as indicated by the range between the green and the black lines in Fig. 8d. Yet, future emissions due to permafrost thaw still depend on the hydrological changes of the landscape."

Line 708: remove "despite"

Responses: We removed the word "despite".

Line 710-712: Not sure I agree. If I understand correctly, then WAD2M shows low wetland extent when soils are frozen. If you are using the mean over several years, then it is not possible to have higher emissions in years where spring thaw comes early and wetland extent in WAD2M would have been higher as a result.

Responses: With the mean seasonal dynamics of WAD2M, we agree that interannual variations in spring emissions from additional wetland areas due to early thaw could be missed. However, the higher emissions in existing wetland areas of WAD2M mean can be modeled due to higher flux rate responses to the earlier temperature increase.

Line 725 and Fig S11: the colors in S11 are very hard to distinguish from each other. Please use a more distinct palette.

Responses: We updated Figure S11 with a more distinct color palette.

Line 731: Which domain?

Responses: We use "domain emission" to represent the study domain, which denotes the wetlands of >45° N.

Line 732: "the percentage of a variation to the period mean of a subregion". Very unclear. What does this mean?

Responses: We revised this sentence:

"The relative interannual variability, which was calculated as the percentage of a subregional variation to its period mean, was attributed to those from West Siberia, Fennoscandia, West Canada, and Alaska (Fig. 10b)."

Figure 10: I don't understand the numbers behind this graph. If the interannual variability is with respect to a mean, then how can the average be non-zero? Or are these boxplots showing medians? With so few years it's better to replot this in a similar style as Fig 9.

Responses: Yes, the lines in the boxes of boxplots show the medians of the data. We added a detailed explanation of the boxplots in the caption:

"The boxplots show the first quartile, the median, and the third quartile of the data with the whiskers denoting the largest variations from the period mean."

Line 761: it's correct that soil temperature is more variable than air temperature, but I don't think that the coarse scale from MERRA can help there since it doesn't model snow cover at the resolution that you need. This goes against your argument in line 765. Perhaps the soil temperatures work better because they have a different amplitude and also perhaps showing a lag to air temperature?

Responses: We agree and express this point with a sentence added as below:

"Although the coarse resolution soil temperatures of MERRA2 may not represent the spatial variability at our upscaling scale, the different amplitudes and lagging responses to air temperatures may improve upscaling models."

Line 792: Which performance requirements?

Responses: We revised this sentence with the requirement details:

"Validation of the SMAP level 4 soil moisture data assimilation product has shown that it meets the performance requirement of unbiased root-mean-square error <0.04 $m^3/m^3$ (Colliander et al., 2022)."

Line 800: Is RF even useful for hysteresis effects?

Responses: The RF model simulated regional hysteresis. However, site-level hysteresis effects were not reconstructed at many sites.

Line 825: "Data deficiency in winter" is this the flux data? Or the limited applicability of WAD2M when soils are frozen?

Responses: We revised this sentence in line xx:

"Data deficiency in EC $CH_4$ flux observations in winter and under-represented areas limited the RF model's extrapolation ability."

Line 880: How are they comparable? What's the difference on an annual basis?

Responses:

In the original manuscript, we used the numbers reported by Rocher-Ros et al., 2023 and Johnson et al., 2022 for estimated annual $CH_4$ emissions from rivers, streams, and lakes in the Arctic and boreal region (>50°N), where the regional numbers were provided in their papers but covered smaller areas than our study region (>45°N) and lake emissions from ice out and spring/fall turnover were not included.

To make a more precise comparison, we downloaded their published spatial data and calculated the annual emissions in areas that exactly match our study region and included emissions from all components of wetlands, rivers/streams, and lakes. We updated the numbers in our manuscript as below:

"$CH_4$ emissions were estimated at 5.5 Tg $CH_4$ yr$^{-1}$ from rivers and streams (Rocher-Ros et al., 2023) and 16.6 Tg $CH_4$ yr$^{-1}$ from lakes (Johnson et al., 2022) in the high latitudes (>45°N). The total emissions estimated from wetlands and open water are about 44.9 Tg $CH_4$ yr$^{-1}$, which is 4 Tg $CH_4$ yr$^{-1}$ more than the CarbonTracker-$CH_4$ estimate."

Supplemental Text 2, line 16-27: please move this to the main document, because it answers a lot of questions that I had on why sites were missing from your analysis. Also, what was the quality control mentioned in line 26?

Responses:

Thanks for your suggestion. We moved the paragraph to the Data section.

The quality control we mentioned in line 26 is described further down in the manuscript:

"Daily data entries were only kept when the number of half-hourly EC tower observations per day was greater than 11."

---

## Author Response (AR2)

We thank the Referee Frans-Jan Parmentier for the helpful comments and suggestions. We addressed comments in the order discussed by the Referee and improved our manuscript accordingly. Our responses are in blue as shown in the supplement document.

**Referee #1**

I'm glad to see the revision by the authors, and that they incorporated all of my comments adequately. I'm also sorry that it took this long to deliver my review of this revision. I think that the paper has become a lot better, and I only have a few minor remaining comments.

Line 233: The exclusion of CA-BOU and RU-COK is based on the QC, but it's never really explained in detail on what exact basis they were excluded. Can you elaborate?

**Responses:**

We communicated with the data manager of the CA-BOU site to help with quality control. We also compared data anomalies in 2018 at the RU-COK site with the data before our study period. We modified line 233-234 to explain that the CA-BOU and RU-COK sites were excluded due to invalid seasonal measurements potentially due to device malfunction:

"Another 2 sites (CA-BOU, RU-COK) were excluded after quality control revealed an instrument anomaly that affected the measurements."

Line 679-680: I'm happy to see this additional analysis, and I know I asked for it, but it would be good if you could add some text for your readers on the reasoning for using maximum annual extent.

**Responses:**

Thanks for the helpful suggestion. We added an explanation on using the maximum annual extent of WAD2M in line 712 - 714:

"The addition of maximum annual wetland extent further constrains the limitations of seasonal WAD2M extents in underestimating methane emitting surface for northern high latitude wetlands, especially in cold seasons."

Line 735-737: This belongs in the discussion, not the results.

**Responses:**

Thanks for your suggestion. This point was elaborated in the Introduction paragraph 3 lines 95-97. We deleted this sentence because it does not belong to the result section. Line 859: The site RU-VRK is not located in the West-Siberian lowlands, it's on the other side of the Ural Mountains.

Responses:

We improved the precision of the sentences in line xxx - xxx:

"For instance, Western Siberian Lowlands, the large wetland complex and the major contributor of interannual variations of CH4 in the region has little data. The nearest site (RU-VRK, not included in this study due to the observations before our study period) is situated on the western side of the Ural mountains, within the Usa River Depression."

Otherwise, I would just like to say that I'm happy to see the vast improvement to this paper, and I think that it'll be a nice addition to the literature.

Frans-Jan Parmentier

We thank the Referee #2 for the helpful comments and suggestions. We addressed comments in the order discussed by the Referee and improved our manuscript accordingly. Our responses are in blue as shown in the supplement document.

The authors put much effort into improving the manuscript and the modeling strategy. Thanks for clarifying the site-level and gridded modeling strategy both in the response letter and the manuscript, making it much easier to understand compared with the first version. However, many parts are still not clear or rigorous enough, making the current manuscript fail to achieve the standard of publishment. Detailed comments are as follows. Major comments:

1. Comment 1: Please clarify in the manuscript that you chose the top 10 most important variables as input variables because the performance converged by R2. Please add model performance results and the converged R2 plot in the manuscript or supplement. Responses:

We chose the top 10 most important variables based on the importance of impurity decrease in random forest modeling as shown in Fig. S3. We also added Fig. S4 and a sentence in the manuscript (line 529-530) as suggested to demonstrate the model performance converged as the increment of input variables by the importance rank.

Fig. S4 Site-level model performance (out-of-bag R2) converged as the increment of predictor variables ordered by the importance rank as shown in Fig. S3. Tick labels on the

x-axis represent the addition of a variable to the precedent variables on the left in a recursive modeling.

2. For modeling strategy, you build site-level models and only use them for variable selection. All the upscaling work is based on the grid-level models. Is my understanding correct? (If I misunderstood, please correct it and clarify it in the manuscript.) Why not directly use the gridded model? You can directly establish gridded models, evaluate feature importance (with all candidate variables), and finally use the several most important variables that achieve the best performance to do the upscaling. I think it would be more reasonable and straightforward, and can help establish the best gridded model for upscaling.

**Responses:**

Yes, all the upscaling work is based on the grid-level models. We modified the objectives in the Introduction section (line 179 - 182) to elaborate this strategy. We agree that directly establishing the best-gridded model is straightforward from the upscaling perspective. However, the antecedent site-level modeling for physical predictor selection helped confirm the site-level controlling factors we learned from the literature and narrow down the gridded candidates. We also compared how the differences in scales and measuring methods between in situ predictors and gridded proxies affect model-learned temporal variability in CH4 fluxes (line 524-528), which could evaluate the impacts of input spatial resolutions on the model performance.

3. I am still curious whether the site-level model can guide the gridded model in feature selection. Actually, site-level and gridded upscaling models are totally different, with different input features (different data sources and different number of features). Comparing Fig S3 and Fig 3b, feature importance differs significantly in site-level and gridded models. For example, TS\_2 showed 1st place in the gridded model, but 8th in the site model. In this situation, can the feature importance by site model still effectively and reasonably guide feature selection for gridded models?

**Responses:**

The site-level model, site synthesis literature (Knox et al., 2021; Delwiche et al., 2021), and expert knowledge guided our predictor variable selection for gridded models. We used the gridded version of those in situ measured physical variables selected at the site-level modeling. Because the data sources were different, we separated the site-level modeling and grid-level modeling to ensure the data sources within a model were comparable. In this way, the feature importance of gridded models can reflect the relative importance of each gridded variable in the gridded-level and upscaling models. This also applied to the site-level modeling. However, the feature importance also pertains to the input data distribution and random forest model structure. Additional gridded variables from remote sensing products were added to complement the missing controllers from the site-level modeling, as you mentioned "different number of features". Therefore, the feature importance by site model can help us identify controlling physical variables but would not necessarily translate to the same rank in the feature importance of grid models. We added sentences in line 389-396 to address this comment.

4. What is the standard for choosing candidate variables in the site model? According to the statements in the main text L268-269, variables that affect CH4 at multi-day to seasonal scales (by previous studies) should be considered. Many studies have revealed considerable impacts of vegetation. For example, in Knox et al. 2021 (you also cited this paper in your manuscript), GPP and RECO significantly controlled CH4 at seasonal scales. Therefore, excluding vegetation proxy in candidate variables for feature selection seems unreasonable. In your response letter, you mentioned excluding vegetation-related variables because neither Peltola nor McNicol used EVI for upscaling, and it seems inconsistent and contradictory with the candidate selection rule you wrote in the main text in L268-269.

**Responses:**

One standard for choosing candidate variables at site-level modeling is that they are in-situ measured physical variables and are available at most sites. We modify lines 268-269 to stress this standard in the main text. We also acknowledge the significant roles that GPP and RECO play in controlling seasonal CH4 changes. GPP and RECO are estimated from measured NEE. The partitioning in GPP and RECO is associated with assumptions that yield unknown amounts of uncertainty. Therefore, we did not include GPP and RECO at site-level modeling but included vegetation-related variables in our grid-level modeling and upscaling. We used MODIS NBAR as proxies for vegetation productivity because EVI and GPP can be derived from MODIS NBAR bands, instead of directly using MODIS-based EVI. We modify line 342-344 in Data section 2.2.2 to clarify this point.

5. L386-389: It is unclear how to select variables for the 'baseline' model. 1) What is the standard for choosing the variables? Did you choose the variables with more 'red' grids and less 'white' grids? 2) In Fig S14, you set a threshold of 0.8. Why use that value? 3) Why choosing ts\_2 instead of ts\_1 or ts\_3? What is the difference? 4) Why can NBAR be excluded from the feature selection based on the pairwise Pearson correlation test?

**Responses:**

We first group significantly correlated variables (p<0.001, r>0.8, white grids except for those on the diagonal line) in the pairwise Pearson correlation test, forming three groups: SMAP soil moisture variables in group 1 (we also include surface soil moisture that is significantly correlated with the other two soil moisture variables and r>0.7), air temperature (tas), downward longwave radiation (rsdl), spfh, soil temperatures (ts1, ts2, and ts3) in group 2, downward shortwave radiation (rsds) and latent heat (le) in group 3. We then select one most important variable in each group according to Fig. S15 for the baseline models. The rest variables out of the groups (air pressure (pa), sensible heat (h), slope, spi, and cti) are included in the baseline models. We added these sentences (line 402-410) to the paragraph about constructing five model settings in section 2.3.1.

6. Figure 3c. Which model did you finally choose to generate the upscaling product? The models with 'all' variables? If so, I think it would be more reasonable to show the feature importance of 'ALL' variables instead of 'baseline' variables. Although collinearity could affect the feature importance, the feature importance of the model with all the variables you finally used for upscaling still should be presented and explained.

**Responses:**

Yes, we chose the models with 'all' variables as they provided the highest mean R2 and lowest median errors (Figure 3a) for all validation sites under the LOOCV scheme. We add the feature importance of 'ALL' variables in the supplementary (Figure S15):

"Fig. S15 Variable importance of "ALL" models with all gridded input variables under the LOOCV scheme: mean (last row) and at each validation site (row denoted by site ID). The 'ALL' modeling setting is used to build the upscaling ensemble models."

We explain the feature importance of 'ALL' variables and the agreement with those of the baseline variables (Figure 3b) in line 571-574:

"The average importance of 'all' gridded variables used for upscaling (Fig. S15) was consistent with baseline models, emphasizing the importance of soil temperatures and rootzone wetness. Additionally, air pressure and topography also contributed to explaining the daily variability in  $CH_4$  fluxes."

**Minor comments:**

1. Fig. S3. I suggest adding the full name of the variables in the caption to make it more readable. Responses: We updated Fig. S3 to label the full names of variables.

2. Fig. S3 and Fig 1: Are the top 10 variables in Fig S3 finally selected? If so, SW\_IN should not be included (it is the 11th). However, in Fig 1, shortwave radiation is included in the models. Responses: We used MERRA2 surface specific humidity as proxies for vapor pressure deficit (VPD) and relative humidity (RH). Therefore, we selected 10 MERRA2 variables as the gridded version of the top 11 variables from site-level modeling.

3. L196-198 'based on literature and expert knowledge': Do you mean you chose these variables based on previous studies? If so, please add citations here. Responses: We add citations for the literature references.

4. L202-203: you mentioned the ML with the highest R2 and lowest ME was chosen. But from Fig 3a, it is hard to tell 'Base+CoVar' or 'All' which is better, they look almost the same in the figure. Which one did you finally use for upscaling? Please clarify it in the manuscript. Responses:

The mean/median R2 of 'All' (0.506/0.55) surpassed those of 'Base+CoVar' (0.505/0.53). The median errors of 'All' (MAE=14.1, RMSE=19.8, ME=4.0, unit: nmol m-2 s-1) were lower than the 'Base+CoVar' (MAE=16.0, RMSE=23.6, ME=6.2, unit: nmol m-2 s-1), while the mean errors of these two modeling settings were comparable. Therefore, we used the 'All' model setting for upscaling. We modified line 202 - 205 in the manuscript to clarify this and added the R2 and error values in line 558-561.

5. Fig S6 is missing in the supplementary. Responses:

Thanks for catching this. We added figures in the supplementary and relabeled the figure numbers. We updated the associated figure numbers in the text of our manuscript.

6. Fig 6. Although you clarified that you used the ensemble mean of bottom-up models in GCP in the manuscript, I suggest using the 'GCP Bottom-up ensemble mean' or 'GCP BU ensemble mean' as the title of the subplot. The 'GCP ensemble mean' can be easily misunderstood as the ensemble mean of all GCP methane models, including both bottom-up and top-down models. The same issue for other figures. Responses:

We updated Fig.6 with your suggested subplot title for 'GCP Bottom-up ensemble mean'. We also updated the captions regarding the bottom-up GCP ensemble mean in other figures.

7. Figure 7d. Why excluding wetCH4\_giems and wetCH4\_GLWD? Responses:

We estimated seasonal emissions with WetCH4\_giems and WetCH4\_GLWD in the North Slope region. GIEMS2 significantly underestimated wetland areas in this region in the summer and estimated zero wetland areas in cold seasons. Fig.7d WetCH4 land (black line) assumed all land in this region as methane emitting surface to represent the potential maximum seasonal CH4 emissions that WetCH4 could estimate. WetCH4 GLWD emissions were between WetCH4 CALU and WetCH4 land. We selected emission estimations with WAD2M and all land that matched the minimum and maximum range of CARVE estimates.

8. If you used microbial emissions instead of wetland emissions for CarbonTracker, would that still be comparable with WetCh4, GCB ensemble, and WetCHARTs? You also mentioned in L662-663 that non-wetland emissions also have considerable contributions, but the current results by Carbon Tracker cannot distinguish them. Responses:

The contributions of ruminant animals and winter open water to the microbial emissions estimated by CarbonTracker-CH4 were negligible in the northern high latitudes (60° - 90° N). When we compared WetCH4, GCP bottom-up ensemble mean, and WetCHARTs in the northern high latitudes, we found an agreement in the phase of the seasonal pattern (Fig. 7b). The differences in the magnitudes in the summer may be attributed to aquatic emissions (Johnson et al., 2022).